# Experimental determination of Henry's law constants of difluoromethane (HFC-32) and the salting-out effects in aqueous salt solutions relevant to seawater

Shuzo Kutsuna

National Institute of Advanced Industrial Science and Technology (AIST), 16-1 Onogawa, Tsukuba, 305-8569, Japan

*Correspondence to*: S. Kutsuna (s-kutsuna@aist.go.jp)

**Abstract.** Gas-to-water equilibrium coefficients, $K_{eq}{}^{S}$ (in M atm$^{-1}$) of difluoromethane ($CH_2F_2$), a hydrofluorocarbon refrigerant (HFC-32), in aqueous salt solutions relevant to seawater were determined over a temperature ($T$) range from 276 to 313 K and a salinity ($S$) range up to 51 ‰ by means of an inert-gas stripping method. From the van't Hoff equation, the $K_{eq}{}^{S}$ value in water, which corresponds to the Henry's law constant ($K_H$), at 298 K was determined to be 0.064 M atm$^{-1}$. The salinity dependence of $K_{eq}{}^{S}$ (the salting-out effect), $\ln(K_H/K_{eq}{}^{S})$, did not obey the Setchenow equation but was proportional to $S^{0.5}$. Overall, the $K_{eq}{}^{S}(T)$ value was expressed by $\ln(K_{eq}{}^{S}(T)) = -49.71 + (77.70 - 0.134 \times S^{0.5}) \times (100/T) + 19.14 \times \ln(T/100)$. By using this equation in a lower tropospheric semi-hemisphere (30° S−90° S) of the Advanced Global Atmospheric Gases Experiment (AGAGE) 12-box model, we estimated that 1 to 4 % of the atmospheric burden of $CH_2F_2$ resided in the ocean mixed layer and that this percentage was at least 4 % in the winter; dissolution of $CH_2F_2$ in the ocean may partially influence estimates of $CH_2F_2$ emissions from long-term observational data of atmospheric $CH_2F_2$ concentrations.

## 1 Introduction

Hydrofluorocarbons (HFCs) have been developed as replacements for chlorofluorocarbons and hydrochlorofluorocarbons (HCFCs) to protect the stratospheric ozone layer from depletion. In particular, difluoromethane (HFC-32, $CH_2F_2$) has been used as a refrigerant to replace HCFC-22 ($CHClF_2$): azeotropic mixtures of $CH_2F_2$ with HFC-125 ($CHF_2CF_3$) and HFC-134a ($CH_2FCF_3$) have been used as refrigerants for air conditioning and refrigeration for a few decades, and $CH_2F_2$ alone has recently been used as a refrigerant for air conditioning.

However, HFCs can act as greenhouse gases, and thus there is concern about emissions of $CH_2F_2$ and other HFCs to the atmosphere, where they can accumulate and contribute to global warming (IPCC, 2013). Observational data from the Advanced Global Atmospheric Gases Experiment (AGAGE) indicate that the atmospheric concentration of $CH_2F_2$ has been increasing every year since 2004; in 2012, the global mean mole fraction of $CH_2F_2$ was 6.2 ± 0.2 parts per trillion (ppt), and the rate of increase was 1.1 ± 0.04 ppt y$^{-1}$ (17% y$^{-1}$) (O'Doherty et al., 2014). By using AGAGE data in combination with a chemical transport model such as the AGAGE 12-box model (Cunnold et al., 1994; Rigby et al., 2013) and a value of 5.1 years as the atmospheric lifetime of $CH_2F_2$, O'Doherty et al. (2014) estimated the global emission rate of $CH_2F_2$ in 2012 to be

21 ± 11 Gg y$^{-1}$ with an increase rate of 14 ± 11% y$^{-1}$. Such estimates on the basis of long-term observational data such as the AGAGE and the National Oceanic and Atmospheric Administration Global Monitoring Division (NOAA GMD) network are called top-down estimates and have been shown to provide an independent and effective method for assessing the accuracy of globally and regionally aggregated reductions or increases in emissions of individual HFCs, as well as other greenhouse gases, compiled from national reports to the United Nations Framework Convention on Climate Change (eg. Prinn et al., 2000; Lunt et al., 2015; Montzka et al., 2015).

The atmospheric lifetimes of HFCs are thus related to their estimated emission rates. The currently accepted value of the atmospheric lifetime of $CH_2F_2$, which was revised in 2014 (Carpenter et al., 2014), is 5.4 years. The partial atmospheric lifetime of $CH_2F_2$ with respect to gas-phase reactions with OH in the troposphere is 5.5 years, and that with respect to stratospheric removal processes is 124 years. Other processes, such as dissolution into seawater, are not considered to contribute significantly to atmospheric removal of $CH_2F_2$. Yvon-Lewis and Butler (2002) estimated partial atmospheric lifetimes of some HCFCs and HFCs with respect to irreversible dissolution into seawater by using physicochemical properties such as solubility and aqueous reaction rates, as well as meteorological data such as temperature and wind speed over the ocean in grids. Their estimates indicated that dissolution into seawater is not a significant sink of the HCFCs and HFCs that were evaluated in the study. Because no aqueous reactions of $CH_2F_2$ have yet been observed under environmental conditions, dissolution of $CH_2F_2$ into seawater is considered to be reversible and cannot serve as a sink of $CH_2F_2$. However, because $CH_2F_2$ is more soluble in water than HCFCs and other HFCs (Sander, 2015), even reversible dissolution of $CH_2F_2$ into seawater might influence a top-down estimate of $CH_2F_2$ emission rates.

The objective of the present study is to experimentally determine the seawater solubility of $CH_2F_2$, which is a physicochemical property necessary for estimating the residence ratio of $CH_2F_2$ in the ocean when the ocean mixed layer is at solubility equilibrium with the atmosphere. Specifically, the Henry's law constants, $K_H$ (in M atm$^{-1}$), of $CH_2F_2$ and the salting-out effects of seawater-relevant ions on $CH_2F_2$ solubility were experimentally determined. Values of $K_H$ for $CH_2F_2$ have been reported in some review papers (Sander, 2015; Clever et al., 2005). The largest and smallest values at 298K differ from each other by a factor of approximately 3: 0.87 M atm$^{-1}$ (Sander, 2015; Yaws and Yang, 1992) and 0.30 M atm$^{-1}$ (Clever et al., 2005; Miguel et al., 2000). To the author's knowledge, no data on the salting-out effects of seawater-relevant ions on $CH_2F_2$ solubility have been reported.

First, the values of $K_H$ for $CH_2F_2$ were determined over the temperature range from 276 to 313 K by means of an inert-gas stripping (IGS) method. The $K_H$ values were also determined over the temperature range from 313 to 353 K by means of a phase ratio variation headspace (PRV-HS) method. The $K_H$ values obtained by the two methods could be fitted by an equation representing the same temperature dependence. Second, salting-out effects on $CH_2F_2$ solubility were determined over the temperature range from 276 to 313 K for test solutions of artificial seawater prepared over the salinity range from 4.5 to 51.5 ‰. The salting-out effects were confirmed for the artificial seawater, but the relationship between $CH_2F_2$ solubility and salinity of the artificial seawater was found to be unusual in that the excessive free energy for dissolution was not proportional to the salinity but rather was represented by an equation involving the salinity and the 0.5 power of the

salinity. Over the salinity range relevant to seawater, the solubility of $CH_2F_2$ in the artificial seawater could be represented as a function of both salinity and temperature. Third, on the basis of the solubility of $CH_2F_2$ in seawater determined in this study and a global gridded dataset of monthly mean values of temperature, salinity, and depth of the ocean mixed layer, the amounts of $CH_2F_2$ dissolved in the ocean mixed layer were estimated in each month for each lower tropospheric semi-
hemisphere of the AGAGE-12 box model.

## 2 Experimental

### 2.1 Materials

$CH_2F_2$ gas (1010 ppmv or 1000 ppmv in synthetic air) was purchased from Takachiho Chemical Industrial Co. (Tokyo, Japan). Sodium chloride (NaCl, >99.5%), sodium sulfate ($Na_2SO_4$, >99%), magnesium chloride ($MgCl_2 \cdot 6H_2O$, >98%),
calcium chloride ($CaCl_2 \cdot 2H_2O$, >99.9%), and potassium chloride (KCl, >99.5%) were purchased from Wako Pure Chemical Industries (Osaka, Japan) and used as supplied. Water was purified with a Milli-Q Gradient A10 system (resistivity > 18 megohm-cm).

Synthetic artificial seawater was prepared as described by Platford (1965) and was used to evaluate the salting-out effects on the solubility of $CH_2F_2$ in the ocean. The prepared artificial seawater had the following definite mole ratios:
0.4240 NaCl, 0.0553 $MgCl_2$, 0.0291 $Na_2SO_4$, 0.0105 $CaCl_2$, and 0.0094 KCl. The ionic strength of the artificial seawater was set between 0.089 and 1.026 mol $kg^{-1}$ water, that is, at molality base, with each salt at the aforementioned mole ratio; the salinity (in ‰) of this artificial seawater was between 4.45 and 51.53‰. This artificial seawater is referred to hereafter as a-seawater.

### 2.2 Inert-gas stripping method with a helical plate

An inert-gas stripping (IGS) method (Mackay et al., 1979) was used to determine the solubility of $CH_2F_2$ in water and aqueous salt solutions. A $CH_2F_2$–air-nitrogen mixture (mixing ratio of $CH_2F_2 \sim 10^{-4}$) was bubbled into the aqueous solution for a certain time period (e.g., 5 min), and then nitrogen gas ($N_2$) was bubbled through the resulting aqueous solution containing $CH_2F_2$, which was stripped from the solution into the gas phase.

The gas-to-liquid partition coefficient (in M $atm^{-1}$) at temperature $T$ (in K), $K_{eq}(T)$, was calculated from the rate of
decrease of the gas-phase partial pressure according to Eqs. (1) and (2):

$$\ln(P_t/P_0) = -k_1 t \tag{1}$$

$$k_1 = \frac{1}{K_{eq}(T)RT} \frac{F}{V} \tag{2}$$

where $P_t/P_0$ is the ratio of the partial pressure of $CH_2F_2$ at time $t$ to the partial pressure of $CH_2F_2$ at fixed time $t_0$; $k_1$ is the first-order decreasing rate constant (in $s^{-1}$); $F$ is the flow rate of $N_2$ (in $dm^3 s^{-1}$); $V$ is the volume of water or aqueous salt

solution (in dm$^3$); and $R$ is the gas constant (0.0821 dm$^3$ atm K$^{-1}$ mol$^{-1}$). The $K_{eq}(T)$ values in water correspond to the Henry's law constants, $K_H(T)$ in M atm$^{-1}$. The $P_t$ values typically ranged from $10^{-4}$ to $10^{-6}$ atm.

A stripping column apparatus with a helical plate was used to strip $CH_2F_2$. This apparatus was described in detail by Kutsuna and Hori (2008) and is described briefly here. The stripping column consisted of a jacketed Duran glass column (4 cm i.d. $\times$ 40 cm height) and a glass gas-introduction tube with a glass helix. Water or a-seawater (0.300 or 0.350 dm$^3$) was added to the column for the test solution. The solution was magnetically stirred, and its temperature was kept constant within $\pm0.2$ K by means of a constant-temperature bath that had both heating and cooling capabilities (NCB-2500, EYELA, Tokyo, Japan) and was connected to the water jacket of the column.

Experiments were conducted at nine temperatures in the range of 276 to 313 K. A $CH_2F_2$–air mixture or $N_2$ was introduced near the bottom of the column through a hole (~1 mm in diameter) in the gas-introduction tube. The bubbles travelled along the underside of the glass helix from the bottom to the top of the column, at which point they entered the headspace of the column. The gas flow was controlled by means of calibrated mass flow controllers (M100 Series, MKS Japan, Inc., Tokyo, Japan) and was varied between $2.2 \times 10^{-4}$ and $4.4 \times 10^{-4}$ dm$^3$ s$^{-1}$ (STP).

The volumetric flow rate of the gas ($F_{meas}$) was calibrated with a soap-bubble meter for each experimental run. The soap-bubble meter had been calibrated by means of a high-precision film flow meter SF-1U with VP-2U (Horiba, Kyoto, Japan). Errors of $F_{meas}$ are within $\pm1\%$. To prevent water evaporation from the stripping column, the gas was humidified prior to entering the stripping column passage through a vessel containing deionized water. This vessel was immersed in a water bath at the same temperature as the stripping column. All volumetric gas flows were corrected to prevailing temperature and pressure by Eq. (3) (Krummen et al., 2000). Errors due to this correction are within $\pm1\%$. Errors of $F$ are thus within $\pm1.4\%$.

$$F = F_{meas} \times \frac{P_{meas} - h_{meas}}{P_{hs} - h} \times \frac{T}{T_{meas}} \tag{3},$$

where $P_{meas}$ and $T_{meas}$ are the ambient pressure and temperature, respectively, at which $F_{meas}$ was calibrated; $P_{hs}$ is the headspace total pressure over the test solution in an IGS method experiment with a flow rate of $F$ at temperature of $T$; and $h_{meas}$ is the saturated vapour pressure, in atm, of water at $T_{meas}$; $h$ is the saturated vapor pressure, in atm, of water or a-seawater at $T$. Values of $h_{meas}$ and $h$ were calculated by use of Eq. (4) where $S$ is salinity of a-seawater (Weiss and Price, 1980).

$$h \text{ or } h_{meas} = \exp\left[24.4543 - 67.4509 \times \left(\frac{100}{T}\right) - 4.8489 \times \ln\left(\frac{T}{100}\right) - 0.000544 \times S\right] \tag{4}$$

The purge gas flow exiting from the stripping column was diluted with constant flow of $N_2$ to prevent water vapour from condensing. The $CH_2F_2$ in the purge gas flow thus diluted was determined by means of gas chromatography–mass spectrometry (GC-MS) on an Agilent GC6890N with 5973inert instrument (Agilent Technologies, Palo Alto, CA). A portion of the purge gas containing $CH_2F_2$ stripped from the test solution was injected into the GC-MS instrument in split mode (split ratio = 1:30) with a six-port sampling valve (VICI AG, Valco International, Schenkon, Switzerland) equipped with a

stainless sampling loop (1.0 cm$^3$). Gas was sampled automatically at intervals of 10 to 11 min during an experimental run (which lasted from 2 to 8 h), depending on the decay rate of the partial pressure of $CH_2F_2$. Peaks due to $CH_2F_2$ were measured in selected-ion mode ($m/z = 33$, $CH_2F^+$). A PoraBOND-Q capillary column (0.32-mm i.d. $\times$ 50-m length, Agilent Technologies) was used to separate $CH_2F_2$. The column temperature was kept at 308 K. Helium was used as the carrier gas. The injection port was kept at 383 K.

If $CH_2F_2$ in the headspace over the test solution is redistributed into the test solution, $k_1$ should be represented by Eq. (5) instead of Eq. (2) (Krummen et al., 2000; Brockbank et al., 2013).

$$k_1 = \frac{F}{K_{eq}(T)RTV + V_{head}} \tag{5},$$

where $V_{head}$ is headspace volume over the test solution. In this study, the values of $V_{head}$ were 0.070 and 0.020 dm$^3$ for $V$ values of 0.300 and 0.350 dm$^3$, respectively. Equations (6) and (7) are derived from Eqs. (2) and (5), respectively:

$$K_{eq}(T) = \frac{1}{k_1 RT}\frac{F}{V} \tag{6},$$

$$K_{eq}(T) = \frac{1}{k_1 RT}\frac{F}{V} - \frac{V_{head}}{RTV} \tag{7}.$$

As described in *Results and discussion* (Sect. 3.1), $CH_2F_2$ in the headspace over the test solution was not expected to be redistributed into the test solution. Hence Eq. (6) was used to deduce $K_{eq}(T)$ from $k_1$. Errors of $T$ are estimated to be within $\pm 0.2$ K. These errors of $T$ may give potential systematic bias of ca. $\pm 2\%$ ($\delta K_{eq}/K_{eq}$) where $\delta K_{eq}$ is error of the value of $K_{eq}$. Errors of $F$ are estimated to be less than 1.4 %, and these errors may give potential systematic bias of less than 1.4 % ($\delta K_{eq}/K_{eq}$). Accordingly, for the IGS methods, values of $K_{eq}$ may have potential systematic bias of ca. $\pm 2\%$.

If redistribution of $CH_2F_2$ in the headspace to the test solution had occurred, the values determined using Eq. (6) would be overestimated. Errors due to this redistribution are always negative values. Ratio of the errors to the $K_{eq}$ values (%) is $\frac{100 k_1 V_{head}}{F}$. Values of this ratio increase as values of $K_{eq}$ decrease. Under the experimental conditions here, this ratio is calculated to be from –2.0 % for water at 3.0 °C to −6.5 % for a-seawater at 51.534‰ and 39.5 °C.

**2.3 Phase ratio variation headspace method**

The $K_H$ values of $CH_2F_2$ in water were also determined by means of the phase ratio variation headspace (PRV-HS) method (Ettre et al., 1993) for comparison with the results obtained by the above-described IGS method. The PRV-HS method experiments were performed over the temperature range from 313 to 353 K at 10 K intervals because the headspace temperature in the equipment used here could not be controlled at less than 313 K. The experimental procedure was the same as that described in detail previously (Kutsuna, 2013) and it is described briefly here.

The determination was carried out by GC-MS on an Agilent GC6890N with 5973inert instrument (Agilent Technologies) equipped with an automatic headspace sampler (HP7694, Agilent Technologies). The headspace samples were slowly and continuously shaken by a mechanical set-up for the headspace equilibration time (1 h; see below), and then the headspace gas (1 cm$^3$) was injected into the gas chromatograph in split mode (split ratio = 1:30). The conditions used for GC-MS were the same as those described in Sect. 2.2.

Headspace samples containing five different amounts of $CH_2F_2$ and six different volumes of water were prepared for an experimental run at each temperature as follows (30 samples total). Volumes ($V_i$) of 1.5, 3.0, 4.5, 6.0, 7.5, and 9.0 cm$^3$ of deionized water were pipetted into six headspace vials with a total volume ($V_0$) of 21.4 cm$^3$ ($V_i/V_0$ = 0.070, 0.140, 0.210, 0.280, 0.350, and 0.421, respectively). Five sets of six headspace vials were prepared and sealed. A prescribed volume ($v_j$) of a standard gas mixture of $CH_2F_2$ and air was added to each set of five vials containing the same volume ($V_i$) of water by means of a gas-tight syringe ($v_j$ = 0.05, 0.10, 0.15, 0.20, or 0.25 cm$^3$). The headspace partial pressure of $CH_2F_2$ thus prepared ranged from $10^{-5}$ to $10^{-6}$ atm.

The time required for equilibration between the headspace and the aqueous solution was determined by analyzing the headspaces over the test samples as a function of time until steady-state conditions were attained. In Fig. S1, the relative signal intensities of the GC-MS peaks for $CH_2F_2$, that is, the ratio of the headspace partial pressure at time $t$ to that at 60 min ($P_t/P_{60}$), are plotted against the time ($t_h$) during which samples were placed in the headspace oven. The plot shows that the peak area did not change after 60 min (Fig. S1). Therefore, the headspace equilibration time was set at 1 h for all the measurements.

If $P_{ij}$ is the equilibrium partial pressure (in atm) of a $CH_2F_2$ sample in a vial with volume $V_0$ (in cm$^3$) containing a volume $V_i$ (in cm$^3$) of water and a volume $v_j$ (in cm$^3$) of a $CH_2F_2$ gas mixture, and if $P_j$ is the equilibrium partial pressure (in atm) of $CH_2F_2$ in a sample containing volume $v_j$ (in cm$^3$) of a $CH_2F_2$ gas mixture without water, then Eq. (8) applies:

$$\frac{P_j V_0}{RT} = K_{eq}(T) P_{ij} V_i + \frac{P_{ij}(V_0 - V_i)}{RT} \tag{8}$$

Because the signal peak area of $CH_2F_2$ ($S_{ij}$) at partial pressure $P_{ij}$ is expected to be proportional to $v_j$ for each set of samples with the same $V_i$, a plot of $S_{ij}$ versus $v_j$ should be a straight line that intercepts the origin:

$$S_{ij} = L_i v_j \tag{9}$$

The slope of the line, $L_i$, corresponds to $S_{ij}$ at $v_j$ = 1.0 cm$^3$. If $L$ is the slope corresponding to $P_i$ at $V_i$ = 0, then

$$\frac{1}{L_i} = \frac{1}{L} + \frac{RTK_{eq}(T)-1}{L}\frac{V_i}{V_0} \tag{10}$$

Plotting $1/L_i$ against $V_i/V_0$ gives an intercept of $1/L$ and a slope of $[RT\ K_{eq}(T) -1]/L$, and $K_{eq}(T)$ can be obtained from these two values. Therefore, $K_{eq}(T)$ can be determined by recording the peak area $S_{ij}$, deriving $L_i$ from a plot of $S_{ij}$ versus $v_j$, and then applying regression analysis to plots of $1/L_i$ versus $V_i/V_0$ with respect to Eq. (10).

Furthermore, values of $K_{eq}(T)$ and errors of them were determined by non-linear fitting of the data of $L_i$ and $V_i/V$ by means of Eq. (11), which was obtained from Eq. (10):

$$L_i = \frac{L}{1+(RTK_{eq}(T)-1)\frac{V_i}{V_0}}$$ (11)

Errors of $T$ are estimated to be within ca. 2 K. These errors of $T$ may give potential systematic bias of ca. ±4% ($\delta K_{eq}/K_{eq}$) at

313 K and ca. ±3% ($\delta K_{eq}/K_{eq}$) at 353 K. Errors of $V_0$ are estimated to be less than 1 %, and these errors may give potential systematic bias of less than 1 % ($\delta K_{eq}/K_{eq}$). Accordingly, for the PRV-HS methods, values of $K_{eq}$ may have potential systematic bias of ca. ±4%.

## 3 Results and discussion

### 3.1 Determination of Henry's law constants

In the IGS method experiment, an aqueous solution was purged with $N_2$ to strip $CH_2F_2$ from the solution into the $N_2$ purge gas flow, and the partial pressure of $CH_2F_2$ ($P_t$) in the $N_2$ purge gas flow decreased with time. Typically it took 20–100 min, depending on the purge gas flow rate and the temperature of the solution, for the decrease to show a first-order time profile. From the first-order time profile of $P_t$ for the following period of 2–7 h, during which $P_t$ typically decreased by 2 orders of magnitude, the first-order decreasing rate constant, $k_1$, was calculated according to Eq. (1). Values of $k_1$ were

obtained at different volumes of deionized water ($V$), various purge gas flow rate ($F$), and various temperatures. Figure S2 shows an example of time profile of $P_t$ and how to calculate the $k_1$ value.

Figure 1 plots values of $F/(k_1RTV)$, the right side of Eq. 6, against $F$ for $V$ values of 0.350 dm$^3$ and 0.300 dm$^3$ at each temperature $T$ (K). Table 1 lists the average values of $F/(k_1RTV)$ for $V$ values of 0.350 and 0.300 dm$^3$ at each temperature. The data with errors being >10% of the data was first excluded. Next, some data were excluded for calculation of the

average so that the remaining data were inside the 2σ range. This procedure was iterated until all the data were inside the 2σ range. The data points thus excluded was only for $V$ values of 0.350 dm$^3$ and the number of them were eight or fewer at each temperature.

As is apparent in Fig. 1 and Table 1, the $F/(k_1RTV)$ values for the two $V$ values (0.350 and 0.300 dm$^3$) agreed at each temperature. This agreement strongly suggests that $K_{eq}(T)$ is represented by Eq. (6) rather than by Eq. (7) because, if Eq. (7)

were applicable, the $K_{eq}(T)$ values calculated for the $V$ value of 0.300 dm$^3$ would be inconsistent with those for the $V$ value of 0.350 dm$^3$: the former would be smaller than the latter by 0.007–0.008 M atm$^{-1}$. Redistribution of $CH_2F_2$ between the headspace and the test solution was probably negligible under the experimental conditions here; hence, values of $K_{eq}(T)$ should be calculated from Eq. (6) rather than Eq. (7).

The above-mentioned agreement also supports the idea that gas-to-water partitioning equilibrium of $CH_2F_2$ was

achieved under the experimental conditions used for the IGS method. As described later, the achievement of gas-to-water partitioning equilibrium was also supported by comparison of these data with $K_{eq}(T)$ values obtained by the PRV-HS method.

Hereafter only values of $F/(k_1 RTV)$ for the $V$ value of 0.350 dm$^3$ are used to deduce $K_{eq}(T)$ values. Because the $K_{eq}(T)$ values in water correspond to the Henry's law constants, $K_H(T)$ in M atm$^{-1}$, $K_H(T)$ is used instead of $K_{eq}(T)$ in this section.

Figure 2 plots the average $K_H$ values for the $V$ value of 0.350 dm$^3$ against $100/T$. Error bars of the data represent both $2\sigma$ for the average and potential systematic bias ($\pm 2\%$). Figure 2 also displays the $K_H(T)$ values obtained by the PRV-HS method. The results of the PRV-HS experiments are described in *Supporting Information* (Fig. S3, Fig. S4 and Table S1). The $K_H$ value obtained by the PRV-HS experiments at each temperature and its error were estimated at 95% confidence level by fitting the two datasets at each temperature (Fig. S4) simultaneously by means of the nonlinear least-squares method with respect to Eq. (11). Error bars of the data by PRV-HS method in Fig. 2 represent both errors at 95% confidence level for the regression and potential systematic bias ($\pm 4\%$).

All the $K_H$ values were regressed with respect to the van't Hoff equation (Eq. (12)) with no weighting (Clarke and Glew, 1965; Weiss, 1970):

$$K_H(T) = \exp\left(a_1 + a_2 \times \left(\frac{100}{T}\right) + a_3 \times \ln\left(\frac{T}{100}\right)\right)$$ (12).

The regression with respect to Eq. (12) gave Eq. (13).

$$\ln\left(K_H(T)\right) = -49.71 + 77.70 \times \left(\frac{100}{T}\right) + 19.14 \times \ln\left(\frac{T}{100}\right)$$ (13).

The square-root-of-variance, that is, standard deviation for each fitting coefficient in Eq. (12) is as follows:

$\delta a_1 = 5.5$; $\delta a_2 = 8.3$; $\delta a_3 = 2.8$.

In Fig. 2, the solid curve was obtained by Eq. (13). The $K_H(T)$ values calculated by Eq. (13) are listed in Table 1. Equation (13) can reproduce the average of $K_H$ values at each temperature within an error of 5%. The dashed lines in Fig. 2 represent 95% confidence limits of the regression for fitting the $K_H(T)$ values by Eq. (12). Taking into consideration errors of the $K_H$ values, the $K_H$ values obtained by the two methods were within the 95% confidence limits of the regression by Eq. (12); this result supports the idea that the values determined by the IGS method and the PRV-HS method were accurate.

The Gibbs free energy for dissolution of $CH_2F_2$ into water at temperature $T$ ($\Delta G_{sol}(T)$) and the enthalpy for dissolution of $CH_2F_2$ into water ($\Delta H_{sol}$) can be deduced from $K_H(T)$ by means of Eqs. (14) and (15):

$$\Delta G_{sol}(T) = \mu_l^\circ(T) - \mu_g^\circ(T) = -RT\ln(K_H(T))$$ (14)

$$\Delta H_{sol}(T) = -R\frac{\partial[\ln(K_H(T))]}{\partial(1/T)}$$ (15)

where $\mu_l^\circ(T)$ is the chemical potential of $CH_2F_2$ under the standard-state conditions at a concentration of 1 M in aqueous solutions at temperature $T$; and $\mu_g^\circ(T)$ is the chemical potential of $CH_2F_2$ under the standard-state conditions at 1 atm of partial pressure in the gas phase at temperature $T$. The $K_H(T)$ and $\Delta H_{sol}(T)$ values at 298.15 K were calculated by means of Eqs. (13) and (15) and are listed in Table 2. $K_H(298.15)$ is represented by $K_H^{298}$ hereafter.

Table 2 also lists literature values of $K_H^{298}$ and $\Delta H_{sol}$ at 298.15 K for $CH_2F_2$ reported in two reviews (Clever et al., 2005; Sander, 2015) and by Anderson (2011); the units of the literature data were converted to M atm$^{-1}$ for $K_H^{298}$ and kJ mol$^{-1}$ for $\Delta H_{sol}$. The $K_H^{298}$ value determined in this study was 7−9% smaller than the values reported by Maaßen (1995), Reichl (1995) and Anderson (2011), whereas the value reported by Yaws and Yang (1992), that reported by Hilal et al. (2008) and that reported by Miguel et al. (2000) were 1.3, 1.4 and 0.47 times, respectively, as large as the value determined here. The absolute value of $\Delta H_{sol}$ at 298.15 K determined here was by 1.4−3.4 kJ mol$^{-1}$ less than the values determined by Maaßen (1995), Reichl (1995), Kühne et al. (2005) and Anderson (2011), whereas it was by 10 kJ mol$^{-1}$ less than the value reported by Miguel et al. (2000).

## 3.2 Determination of salting-out effects in artificial seawater

The solubility of $CH_2F_2$ in a-seawater (Sect. 2.1) was determined by means of the IGS method (Sect. 2.2). According to Eq. (6), the $K_{eq}(T)$ values at an a-seawater salinity of $S$ in ‰ were obtained by averaging the $F/(k_1RTV)$ values for the $V$ value of 0.350 dm$^3$ at each salinity and temperature in a similar way as described in Sect. 3.1. Figure 3 plots values of $F/(k_1RTV)$ at each temperature against $F$ for $V$ values of 0.350 dm$^3$ at an a-seawater salinity of 36.074‰. Figures S5-S8 represent such plots at an a-seawater salinity of 4.452, 8.921, 21.520 and 51.534 ‰. The $K_{eq}(T)$ value at an a-seawater salinity of $S$ in ‰ is represented by $K_{eq}^S(T)$ hereafter. Table 3 lists the $K_{eq}^S(T)$ values.

Figure 4 plots the $K_{eq}^S(T)$ values against $100/T$. The plots indicate a clear salting-out effect on $CH_2F_2$ solubility in a-seawater: that is, the solubility of $CH_2F_2$ in a-seawater decreased with increasing a-seawater salinity. For example, the solubility of $CH_2F_2$ in a-seawater at a salinity of 36.074‰ was 0.73–0.79 times the solubility in water at 3.0 to 39.5 °C.

In general, the salting-out effect on nonelectrolyte solubility in an aqueous salt solution of ionic strength $I$ can be determined empirically by means of the Setchenow equation:

$$\ln(K_H(T)/K_{eq}^I(T)) = k_I I \tag{16}$$

where $K_{eq}^I(T)$ is the $K_{eq}(T)$ at ionic strength $I$ in mol kg$^{-1}$; and $k_I$ is the Setchenow coefficient for the molality- and natural logarithm-based Setchenow equation and is independent of $I$ (Clegg and Whitfield, 1991). For a-seawater, a similar relationship between $K_{eq}^S(T)$ and $S$ is expected:

$$\ln(K_H(T)/K_{eq}^S(T)) = k_S S \tag{17}$$

where $k_S$ is the Setchenow coefficient for the salinity- and natural logarithm-based Setchenow equation and is independent of $S$. Figure 5 plots $\ln(K_H(T)/K_{eq}^S(T))$ against $S$ at each temperature. Table S2 lists values of $k_s$ determined by fitting the data at each temperature by use of Eq. (17). If the $K_{eq}^S(T)$ values obeyed Eq. (17), the data at each temperature in Fig. 5 would fall on a straight line passing through the origin, but they did not. Figure 5 reveals that the salinity dependence of $CH_2F_2$ solubility in a-seawater cannot be represented by Eq. (17).

When the same data were plotted on a log–log graph (Fig. S9), a line with a slope of about 0.5 was obtained by linear regression. This result suggests that $\ln(K_H(T)/K_{eq}{}^S(T))$ varied according to Eq. (18):

$$\ln\left(K_H(T)/K_{eq}{}^S(T)\right) = k_{s1} \times S^{0.5} \tag{18}$$

Values of $k_{s1}$ may be represented by the following function of $T$:

$$k_{s1} = b_1 + b_2 \times \left(\frac{100}{T}\right) \tag{19}$$

Parameterizations of $b_1$ and $b_2$ obtained by fitting all the $\ln(K_H(T)/K_{eq}{}^S(T))$ and $S$ data at each temperature simultaneously by means of the nonlinear least-squares method gives Eq. (20).

$$\ln\left(K_H(T)/K_{eq}{}^S(T)\right) = \left(0.0127 + 0.0099 \times \left(\frac{100}{T}\right)\right) \times S^{0.5} \tag{20}$$

The standard deviation for each fitting coefficient in Eq. (19) is as follows:

$\delta b_1 = 0.0106$; $\delta b_2 = 0.0031$.

Since $2 \times \delta b_1 > b_1$, the parameterization by Eq. (19) may be overworked. Accordingly, all the $\ln(K_H(T)/K_{eq}{}^S(T))$ and $S$ data at each temperature are fitted simultaneously using Eq. (21) instead of Eq. (19). The nonlinear least-squares method gives Eq. (22).

$$k_{s1} = b_2 \times \left(\frac{100}{T}\right) \tag{21}$$

$$\ln\left(K_H(T)/K_{eq}{}^S(T)\right) = 0.1343 \times \left(\frac{100}{T}\right) \times S^{0.5} \tag{22}$$

The standard deviation for the fitting coefficient in Eq. (21) is as follows: $\delta b_2 = 0.0013$. As seen in Fig. 5, Eqs. (21) and (22) reproduced the data well.

$\ln(K_H(T)/K_{eq}{}^S(T))$ depends on $S^{0.5}$ and follows Eq. (22) rather than the Setchenow dependence (Eq. (17)). Table S7 compares values of $K_{eq}{}^S$ calculated by Eq. (22) with those by Eq. (17). The difference between these values of $K_{eq}{}^S$ at 35‰ of salinity was within 3% of the $K_{eq}{}^S$ value. Decreases in values of $K_{eq}{}^S$ are calculated to be 7–8% and 4%, respectively, by Eqs. (17) and (23) as salinity of a-seawater increases from 30‰ to 40‰ at each temperature.

The reason for this salting-out effect of $CH_2F_2$ solubility in a-seawater is not clear. Specific properties of $CH_2F_2$ – small molecular volume, which results in small work of cavity creation (Graziano, 2004; 2008), and large solute-solvent attractive potential energy in water and a-seawater− may cause deviation from Setchenow relationship (*Supporting Information*).

In Eq. (22), $K_H(T)$ is represented by Eq. (13), as described in Sect. 3.1. Therefore $K_{eq}{}^S(T)$ is represented by Eq. (23):.

$$\ln\left(K_{eq}{}^S\right) = -49.71 + (77.70 - 0.1343 \times S^{0.5}) \times \left(\frac{100}{T}\right) + 19.14 \times \ln\left(\frac{T}{100}\right) \tag{23}$$

The values calculated with Eq. (23) are indicated by the bold curves in Fig. 4 and are listed in Table 3. Table 3 lists errors at 95% confidence level for the regression. These errors (error$_{23}$) are calculated by Eq. (24):

$$\text{error}_{23} = K_{eq}{}^S \times \sqrt{\left(\frac{\text{error}_{13}}{K_H}\right)^2 + \left(\frac{\text{error}_{22}}{K_{eq}{}^S}\right)^2} \tag{24}$$

where $\text{error}_{13}$ represents errors at 95% confidence level for the regression by Eq. (12); $\text{error}_{22}$ represents errors at 95% confidence level for the regression by Eq. (21). Table 3 also lists errors due to both errors at 95% confidence level for the regression and potential systematic bias ($\pm 2\%$). Equation (23) reproduced the experimentally determined values of $K_H(T)$ and

$K_{eq}{}^S(T)$ within the uncertainty of these experimental runs.

### 3.3 Dissolution of $CH_2F_2$ in the ocean mixed layer and its influence on estimates of $CH_2F_2$ emissions

The solubility of $CH_2F_2$ in a-seawater can be represented as a function of temperature and salinity relevant to the ocean (Eq. (23)). Monthly averaged equilibrium fractionation values of $CH_2F_2$ between the atmosphere and the ocean ($R_m$ in Gg patm$^{-1}$, where patm is $10^{-12}$ atm) in that the ocean mixed layer is at solubility equilibrium with the atmosphere is estimated

as follows. If we divide the global ocean into $0.25° \times 0.25°$ grids, $R_m$ can be estimated from the sum of the equilibrium fractionation values from the gridded cells:

$$R_m = \frac{m_{d,m}}{P_a} = Q \sum_{i=-360}^{i=360} \sum_{j=-720}^{j=720} K_{eq}^S(T) d_{i,j,m} A_{i,j,m} \tag{25}$$

where $m_{d,m}$, in Gg, is the amount of $CH_2F_2$ dissolved in the ocean mixed layer; $p_a$, in $10^{-12}$ atm, is the $CH_2F_2$ partial pressure in the air; $d_{i,j,m}$ is the monthly mean depth, in m, of the ocean mixed layer in each grid cell; $A_{i,j,m}$ is the oceanic area, in m$^2$, in

each grid cell; $Q$ is a conversion factor (with a value of 52); $m$ is the month index; and $i$ and $j$ are the latitude and longitude indices. We obtained monthly $0.25° \times 0.25°$ gridded sea surface temperatures and sea surface salinities from WOA V2 2013 data collected at 10 m depth from 2005 to 2012 (https://www.nodc.noaa.gov/OC5/woa13/woa13data.html; Boyer et al., 2013) and monthly $2° \times 2°$ gridded mean depths of the ocean mixed layer from Mixed layer depth climatology and other related ocean variables in temperature with a fixed threshold criterion (0.2°C)

(http://www.ifremer.fr/cerweb/deboyer/mld/Surface_Mixed_Layer_Depth.php; de Boyer Montégut et al., 2004). Values of $A_{i,j,m}$ were estimated to be equal to the area of each grid cell in which both gridded data were unmasked.

Figure 6 shows the $R_m$ values for the global and the semi-hemispheric atmosphere. Values of $R_m$ for the global atmosphere are between 0.057 and 0.096 Gg patm$^{-1}$. Because $10^{-12}$ atm of $CH_2F_2$ in the global atmosphere corresponds to 9.4 Gg of atmospheric burden of $CH_2F_2$, 0.6 to 1.0 % of the atmospheric burden resides in the ocean mixed layer when that

layer is at solubility equilibrium with the atmosphere. The magnitude of "buffering" of the atmospheric burden of $CH_2F_2$ by the additional $CH_2F_2$ in ocean surface waters is therefore realistically limited to only about 1 % globally. However, such buffering would be more effective in each lower tropospheric semi-hemisphere of the AGAGE 12-box model, which has been used for a top-down estimate of $CH_2F_2$ emissions. The right vertical axis of Fig. 6 represents the residence ratios of $CH_2F_2$ dissolved in the ocean mixed layer for each lower tropospheric semi-hemispheric atmosphere of the AGAGE 12-box

model. The residence ratios were calculated on the assumption that $10^{-12}$ atm of $CH_2F_2$ corresponds to 1.2 Gg of atmospheric

burden of $CH_2F_2$ in each lower tropospheric semi-hemisphere. As seen in Figure 6, in the southern semi-hemispheric lower troposphere (30° S–90° S), at least 5 % of the atmospheric burden of $CH_2F_2$ would reside in the ocean mixed layer in the winter, and the annual variance of the $CH_2F_2$ residence ratio would be 4%. These ratios are, in fact, upper limits because $CH_2F_2$ in the ocean mixed layer may be undersaturated. It takes days to a few weeks after a change in temperature or

salinity for oceanic surface mixed layers to come to equilibrium with the present atmosphere, and equilibration time increases with depth of the surface mixed layer (Fine, 2011). In the estimation using the gridded data here, >90 % of $CH_2F_2$ in the ocean mixed layer would reside in less than 300 m depth (Tables S3, S4, S5 and S6).

     Haine and Richards (1995) demonstrated that seasonal variation in ocean mixed layer depth was the key process which affected undersaturation and supersaturation of chlorofluorocarbon 11 (CFC-11), CFC-12 and CFC-113 by use

of a one-dimensional slab mixed model. As described above, >90 % of $CH_2F_2$ in the ocean mixed layer is expected to reside in less than 300 m depth. According to the model calculation results by Haine and Richards (1995), saturation of $CH_2F_2$ would be >0.9 for the ocean mixed layer with less than 300 m depth. The saturation of $CH_2F_2$ in the ocean mixed layer is thus estimated to be at least 0.8. In the southern semi-hemispheric lower troposphere (30° S–90° S), therefore, at least 4 % of the atmospheric burden of $CH_2F_2$ would reside in the ocean mixed layer in the winter, and the

annual variance of the $CH_2F_2$ residence ratio would be 3%.

     In the Southern Hemisphere, $CH_2F_2$ emission rates are much lower than in the Northern Hemisphere. Hence, dissolution of $CH_2F_2$ in the ocean, even if dissolution is reversible, may influence estimates of $CH_2F_2$ emissions derived from long-term observational data on atmospheric concentrations of $CH_2F_2$; in particular, consideration of dissolution of $CH_2F_2$ in the ocean may affect estimates of $CH_2F_2$ emissions in the Southern Hemisphere and their seasonal variability because of slow rates of

inter-hemispheric transport and small portion of the $CH_2F_2$ emissions in the Southern Hemisphere to the total emissions.

## 4 Conclusion

     The solubility of $CH_2F_2$ in aqueous salt solutions relevant to seawater can be represented as a function of temperature and salinity, as shown in Eq. (22). The relationship between $CH_2F_2$ solubility and the salinity of the artificial seawater was

found to be unusual in that the excessive free energy for dissolution depended predominantly on the 0.5 power of salinity. By using the solubility of $CH_2F_2$ determined in this study, the magnitude of buffering of the atmospheric burden of $CH_2F_2$ by the additional $CH_2F_2$ in ocean surface waters is estimated to be realistically limited to only about 1 % globally; however, in a southern semi-hemispheric lower troposphere (30° S–90° S) of the AGAGE 12-box model, the atmospheric burden of $CH_2F_2$ is estimated to reside in the ocean mixed layer by at least 4 % in the winter and by 1 % in the summer. Hence, it may be

necessary that dissolution of $CH_2F_2$ in the ocean be taken into consideration to derive $CH_2F_2$ emissions in the Southern Hemisphere and their seasonal variability from long-term observational data on atmospheric concentrations of $CH_2F_2$.

**Supplement link**

Supporting information is attached.

**Competing interests**

The authors declare that they have no conflict of interest.

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

**Table 1.** The average of values of $F/(k_1RTV)$ obtained for $V$ value of 0.350 dm$^3$ and 0.300 dm$^3$ and the $K_H(T)$ value derived from Eq. (13) at each temperature. $N$ represents number of experimental runs for the average.

| $T$ (K) | $F / (k_1RTV)$ | | | | $K_H(T)$ (M atm$^{-1}$) |
| | $V = 0.350$ | | $V = 0.300$ | | |
| | average [a, b] | $N$ [c] | average [a] | $N$ [c] | From Eq. (13) [d, e] |
|---|---|---|---|---|---|
| 276.15 | 0.119 ± 0.006 (0.008) | 21 (2) | 0.117 ± 0.006 (0.008) | 11 (0) | 0.118 ± 0.003 (0.005) |
| 278.35 | 0.107 ± 0.005 (0.007) | 18 (3) | 0.110 ± 0.005 (0.007) | 14 (0) | 0.108 ± 0.002 (0.004) |
| 283.65 | 0.093 ± 0.003 (0.005) | 27 (5) | 0.092 ± 0.001 (0.003) | 5 (0) | 0.094 ± 0.002 (0.004) |
| 288.65 | 0.082 ± 0.006 (0.008) | 41 (5) | 0.084 ± 0.006 (0.008) | 12 (0) | 0.082 ± 0.002 (0.004) |
| 293.45 | 0.071 ± 0.001 (0.002) | 15 (8) | 0.071 ± 0.001 (0.002) | 5 (0) | 0.072 ± 0.002 (0.003) |
| 298.15 | 0.064 ± 0.002 (0.003) | 30 (6) | 0.067 ± 0.005 (0.006) | 12 (0) | 0.064 ± 0.002 (0.003) |
| 303.05 | 0.057 ± 0.003 (0.004) | 16 (0) | 0.056 ± 0.005 (0.006) | 4 (0) | 0.058 ± 0.002 (0.003) |
| 307.95 | 0.051 ± 0.001 (0.002) | 12 (6) | 0.054 ± 0.004 (0.005) | 10 (0) | 0.052 ± 0.002 (0.003) |
| 312.65 | 0.046 ± 0.001 (0.002) | 13 (3) | 0.047 ± 0.001 (0.002) | 4 (0) | 0.048 ± 0.001 (0.002) |

a. Errors are $2\sigma$ for the average only.; b. Number in parenthesis represents an error reflecting both $2\sigma$ for the average and potential systematic bias (±2%).; c. Number in parenthesis represents number of experimental runs excluded for the average.; d. Errors are 95% confidence level for the regression only.; e. Number in parenthesis represents an error reflecting both errors at 95% confidence level for the regression and potential systematic bias (±2%).

**Table 2.** $K_H^{298}$ and $\Delta H_{sol}$ values derived from Eqs. (11′) and (13), along with literature data for $K_H^{298}$ and $\Delta H_{sol}$

| $K_H^{298}$ (M atm$^{-1}$) | $\Delta H_{sol}$ (kJ mol$^{-1}$) | |
|---|---|---|
| 0.064 | −17.2 | This work |
| 0.070 | −20 | Maaβen (1995)[a] |
| 0.070 | −19 | Reichl (1995)[a] |
| 0.069[c] | −20.6 | Anderson (2011) |
| 0.085 | | Hilal et al. (2008)[a] |
| | −18.6, −19.7 | Kühne et al. (2005)[a] |
| 0.087 | | Yaws (1999)[a] |
| 0.087 | | Yaws and Yang (1992)[a] |
| 0.030 | −27.2 | Miguel et al. (2000)[b] |

[a] Reviewed by Sander (2015); [b] Reviewed by Clever et al. (2005)

[c] The value was obtained by extrapolation of the data reported at 284.15-296.15 K (Supplementary data in Anderson (2011)) with respect to the van't Hoff equation.

**Table 3.** The average of values of $F/(k_1RTV)$ obtained for $V$ value of 0.350 dm$^3$ and the $K_{eq}^S(T)$ value derived from Eq. (23) at each salinity and temperature. $N$ represents number of experimental runs for the average.

| $T$ (K) | $K_{eq}^S$ (M atm$^{-1}$) | | | | | |
|---|---|---|---|---|---|---|
| | salinity, 4.452 ‰ | | | salinity, 8.921 ‰ | | |
| | average [a, b] | $N$ [c] | Eq. (23) [d, e] | average [a, b] | $N$ [c] | Eq. (23) [d, e] |
| 276.15 | 0.108 ± 0.006 (0.008) | 8 (0) | 0.107 ± 0.003 (0.005) | 0.103 ± 0.006 (0.008) | 21 (0) | 0.102 ± 0.003 (0.005) |
| 278.35 | 0.099 ± 0.004 (0.006) | 12 (0) | 0.098 ± 0.002 (0.005) | 0.095 ± 0.006 (0.008) | 26 (1) | 0.094 ± 0.002 (0.004) |
| 283.65 | 0.086 ± 0.003 (0.005) | 9 (0) | 0.085 ± 0.002 (0.004) | 0.083 ± 0.007 (0.009) | 24 (0) | 0.081 ± 0.002 (0.004) |
| 288.65 | 0.075 ± 0.004 (0.006) | 12 (0) | 0.074 ± 0.002 (0.003) | 0.072 ± 0.005 (0.006) | 33 (0) | 0.071 ± 0.001 (0.002) |
| 293.45 | 0.065 ± 0.002 (0.003) | 10 (0) | 0.065 ± 0.002 (0.003) | 0.063 ± 0.003 (0.004) | 27 (5) | 0.063 ± 0.002 (0.003) |
| 298.15 | 0.058 ± 0.002 (0.003) | 13 (0) | 0.058 ± 0.002 (0.003) | 0.056 ± 0.004 (0.005) | 26 (2) | 0.056 ± 0.002 (0.003) |
| 303.05 | 0.052 ± 0.001 (0.002) | 8 (0) | 0.052 ± 0.002 (0.003) | 0.049 ± 0.004 (0.005) | 14 (6) | 0.050 ± 0.001 (0.002) |
| 307.95 | 0.047 ± 0.002 (0.003) | 13 (1) | 0.048 ± 0.001 (0.002) | 0.046 ± 0.004 (0.005) | 23 (1) | 0.046 ± 0.001 (0.002) |
| 312.65 | 0.042 ± 0.001 (0.002) | 7 (0) | 0.044 ± 0.001 (0.002) | 0.040 ± 0.003 (0.004) | 12 (8) | 0.042 ± 0.001 (0.002) |

| $T$ (K) | $K_{eq}^S$ (M atm$^{-1}$) | | | | | |
|---|---|---|---|---|---|---|
| | salinity, 21.520 ‰ | | | salinity, 36.074 ‰ | | |
| | average [a, b] | $N$ [c] | Eq. (23) [d, e] | average [a, b] | $N$ [c] | Eq. (23) [d, e] |
| 276.15 | 0.095 ± 0.006 (0.008) | 20 (0) | 0.094 ± 0.003 (0.005) | 0.088 ± 0.005 (0.007) | 21 (0) | 0.088 ± 0.002 (0.004) |
| 278.35 | 0.087 ± 0.005 (0.007) | 22 (0) | 0.086 ± 0.002 (0.004) | 0.079 ± 0.006 (0.008) | 20 (3) | 0.081 ± 0.002 (0.004) |
| 283.65 | 0.075 ± 0.004 (0.006) | 15 (1) | 0.075 ± 0.001 (0.003) | 0.069 ± 0.002 (0.003) | 18 (2) | 0.070 ± 0.001 (0.002) |
| 288.65 | 0.066 ± 0.004 (0.005) | 20 (0) | 0.066 ± 0.001 (0.002) | 0.062 ± 0.004 (0.005) | 19 (4) | 0.061 ± 0.001 (0.002) |
| 293.45 | 0.058 ± 0.003 (0.004) | 14 (0) | 0.058 ± 0.001 (0.002) | 0.054 ± 0.002 (0.003) | 19 (4) | 0.055 ± 0.001 (0.002) |
| 298.15 | 0.052 ± 0.003 (0.004) | 20 (0) | 0.052 ± 0.001 (0.002) | 0.049 ± 0.002 (0.003) | 24 (4) | 0.049 ± 0.001 (0.002) |
| 303.05 | 0.046 ± 0.003 (0.004) | 16 (0) | 0.047 ± 0.001 (0.002) | 0.044 ± 0.002 (0.003) | 16 (0) | 0.044 ± 0.001 (0.002) |
| 307.95 | 0.042 ± 0.003 (0.004) | 16 (0) | 0.042 ± 0.001 (0.002) | 0.040 ± 0.002 (0.003) | 15 (2) | 0.040 ± 0.001 (0.002) |
| 312.65 | 0.038 ± 0.002 (0.003) | 16 (0) | 0.039 ± 0.001 (0.002) | 0.036 ± 0.002 (0.003) | 16 (0) | 0.037 ± 0.001 (0.002) |

| $T$ (K) | $K_{eq}^S$ (M atm$^{-1}$) | | |
|---|---|---|---|
| | salinity, 51.534 ‰ | | |
| | average [a, b] | $N$ [c] | Eq. (23) [d, e] |
| 276.15 | 0.081 ± 0.003 (0.005) | 10 (0) | 0.083 ± 0.002 (0.004) |
| 278.35 | 0.077 ± 0.003 (0.005) | 15 (0) | 0.076 ± 0.002 (0.004) |
| 283.65 | 0.067 ± 0.001 (0.003) | 9 (1) | 0.067 ± 0.001 (0.002) |
| 288.65 | 0.059 ± 0.002 (0.003) | 14 (1) | 0.058 ± 0.001 (0.002) |
| 293.45 | 0.052 ± 0.001 (0.002) | 7 (3) | 0.052 ± 0.001 (0.002) |
| 298.15 | 0.047 ± 0.002 (0.003) | 15 (0) | 0.046 ± 0.001 (0.002) |
| 303.05 | 0.042 ± 0.001 (0.002) | 8 (0) | 0.042 ± 0.001 (0.002) |
| 307.95 | 0.038 ± 0.002 (0.003) | 12 (0) | 0.038 ± 0.001 (0.002) |
| 312.65 | 0.036 ± 0.001 (0.002) | 7 (1) | 0.035 ± 0.001 (0.002) |

5   a. Errors are 2σ for the average only.; b. Number in parenthesis represents an error reflecting both 2σ for the average and potential systematic bias.; c. Number in parenthesis represents number of experimental runs excluded for the average.; d. Errors are 95% confidence level for the regression only.; e. Number in parenthesis represents an error reflecting both errors at 95% confidence level for the regression and potential systematic bias (±2%).

**Figure captions**

Figure 1. Plots of values of $F/(k_1RTV)$ against $F$ at each temperature for 0.350 dm$^3$ and 0.300 dm$^3$ of deionized water. Error bars represent 2σ due to errors of values of $k_1$ as described in Sect. S2 in Supporting Information. Grey symbols represent the data excluded for calculating the average.

Figure 2. van't Hoff plot of the $K_H$ values obtained by the IGS method and the PRV-HS method. Bold curve displays the fitting of the data obtained by the IGS method and the PRV-HS method (Eq. (13)). Dashed curves display upper and lower 95% confidence limit of the above fitting by Eq. (12). Error bars of the data by the IGS method represent both 2σ for the average and potential systematic bias (±2%). Error bars of the data by PRV-HS method represent both errors at 95% confidence level for the regression and potential systematic bias (±4%).

Figure 3. Plots of values of $F/(k_1RTV)$ against $F$ at each temperature for 0.350 dm$^3$ of a-seawater at 36.074‰. Error bars represent 2σ due to errors of values of $k_1$ as described in Sect. S2 in Supporting Information. Grey symbols represent the data excluded for calculating the average.

Figure 4. van't Hoff plot of the $K_{eq}^S$ values for a-seawater at each salinity. Dashed curve represents the $K_H$ values by Eq. (13). Bold curves represent the fitting obtained by Eq. (23). Error bars of the data represent both 2σ for the average and potential systematic bias (2%).

systematic bias (2%).

Figure 5. Plots of $\ln(K_H(T)/K_H^S(T))$ vs. salinity in a-seawater at each temperature. Bold curves represent the fitting obtained by Eq. (22). Error bars represent errors reflecting both 2σ for the average and potential systematic bias (2%) of $K_{eq}^S$.

Figure 6. Plots of monthly averaged equilibrium fractionation of $CH_2F_2$ between atmosphere and ocean, $R_m$ (Gg patm$^{-1}$) in the global and the hemispheric atmosphere. Right vertical axis represents the residence ratio of $CH_2F_2$ in the ocean, instead

of $R_m$, for each lower tropospheric semi-hemisphere of the AGAGE 12-box model.

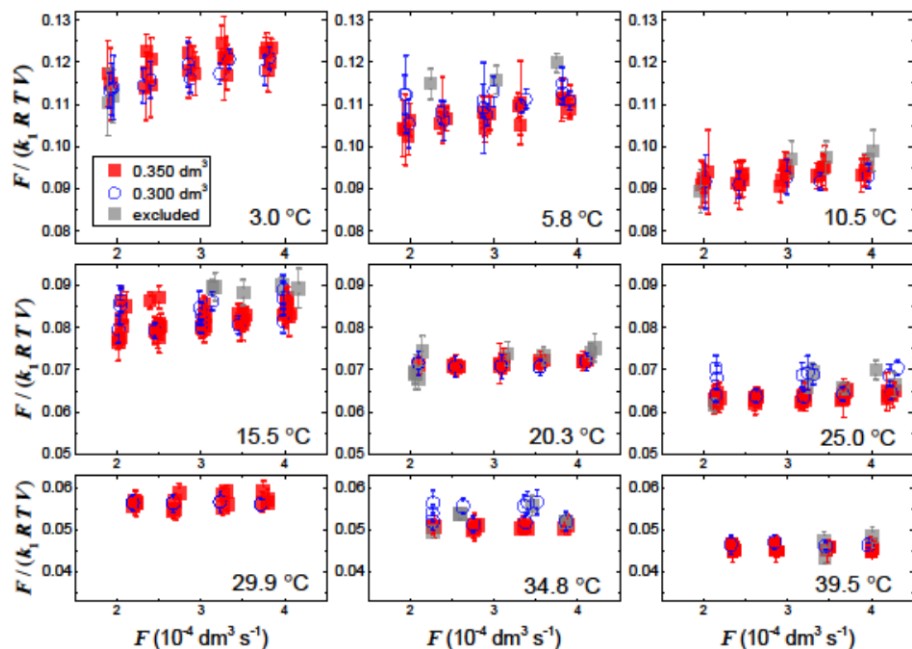

**Figure 1.** Plots of values of $F/(k_1RTV)$ against $F$ at each temperature for 0.350 dm$^3$ and 0.300 dm$^3$ of deionized water. Error bars represent $2\sigma$ due to errors of values of $k_1$ as described in Sect. S2 in Supporting Information. Grey symbols represent the data excluded for calculating the average.

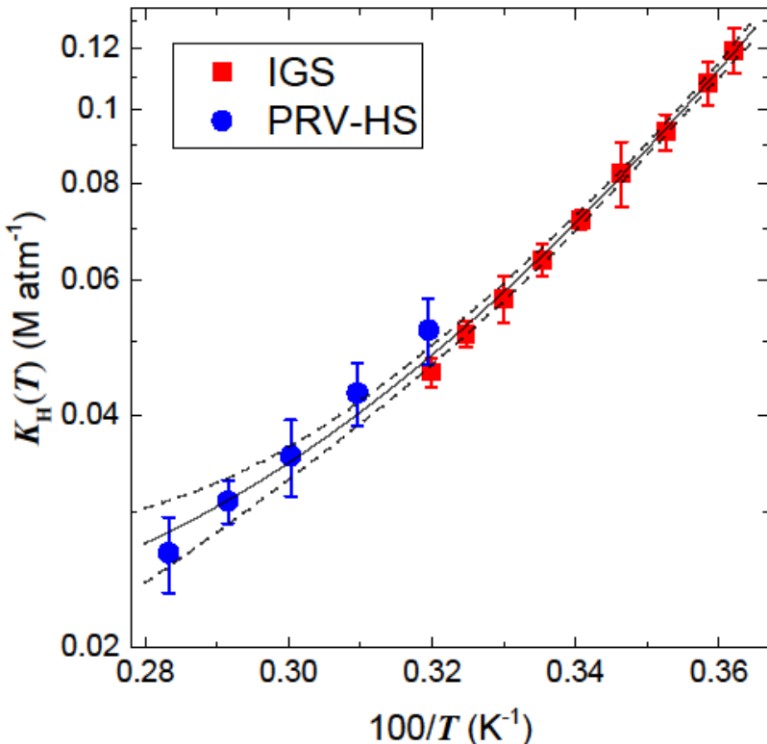

**Figure 2. van't Hoff plot of the $K_H$ values obtained by the IGS method and the PRV-HS method. Bold curve displays the fitting of the data obtained by the IGS method and the PRV-HS method (Eq. (13)). Dashed curves display upper and lower 95% confidence limit of the above fitting by Eq. (12). Error bars of the data by the IGS method represent both 2σ for the average and potential systematic bias (±2%). Error bars of the data by PRV-HS method represent both errors at 95% confidence level for the regression and potential systematic bias (±4%).**

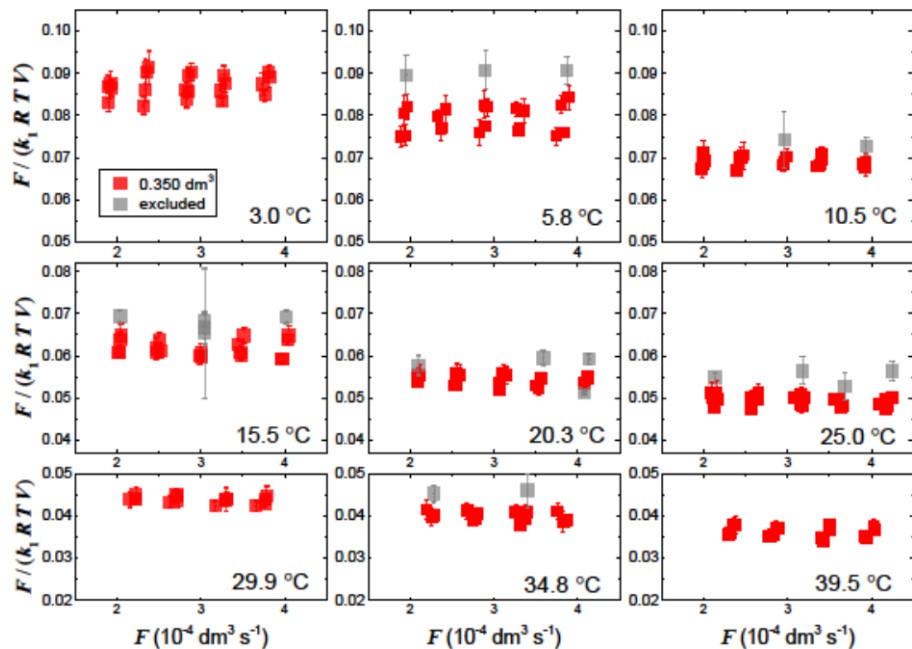

**Figure 3.** Plots of values of $F/(k_1RTV)$ against $F$ at each temperature for 0.35 dm$^3$ of a-seawater at 36.074‰. Error bars represent $2\sigma$ due to errors of values of $k_1$ as described in Sect. S2 in Supporting Information. Grey symbols represent the data excluded for calculating the average.

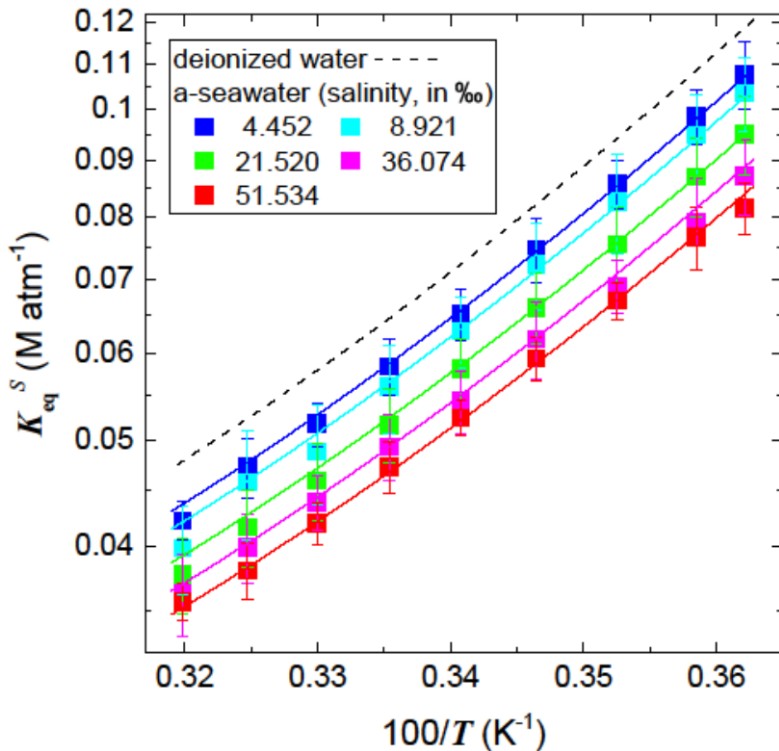

**Figure 4. van't Hoff plot of the $K_{eq}^{S}$ values for a-seawater at each salinity. Dashed curve represents the $K_H$ values by Eq. (13). Bold curves represent the fitting obtained by Eq. (23). Error bars of the data represent both 2σ for the average and potential systematic bias (2%).**

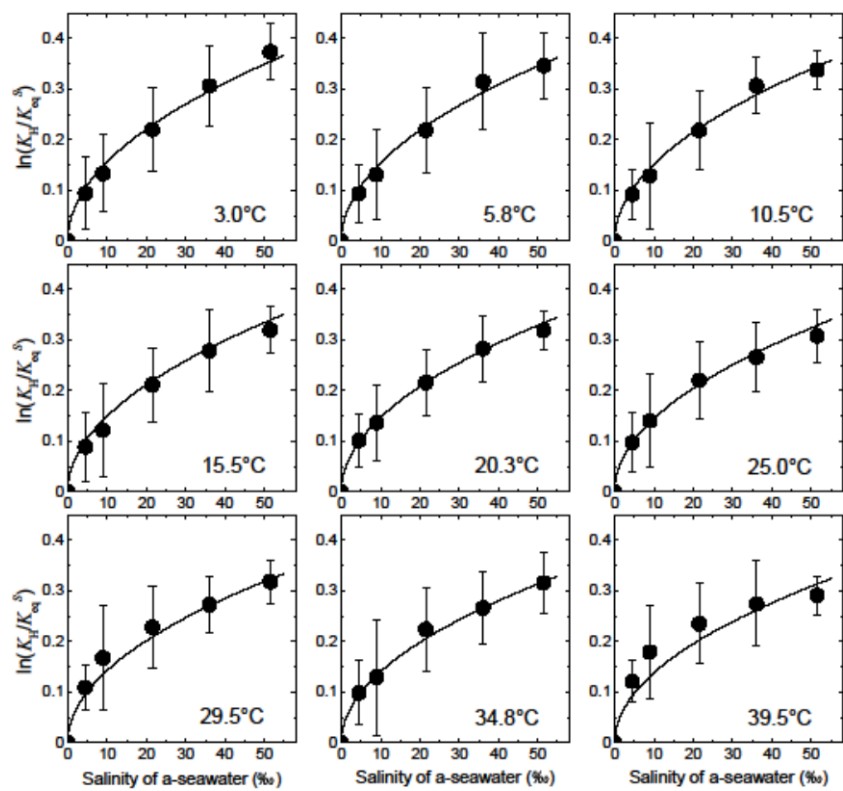

**Figure 5.** Plots of $\ln(K_H(T)/K_H^S(T))$ vs. salinity in a-seawater at each temperature. Bold curves represent the fitting obtained by Eq. (22). Error bars represent errors reflecting both $2\sigma$ for the average and potential systematic bias of $K_{eq}^S$.

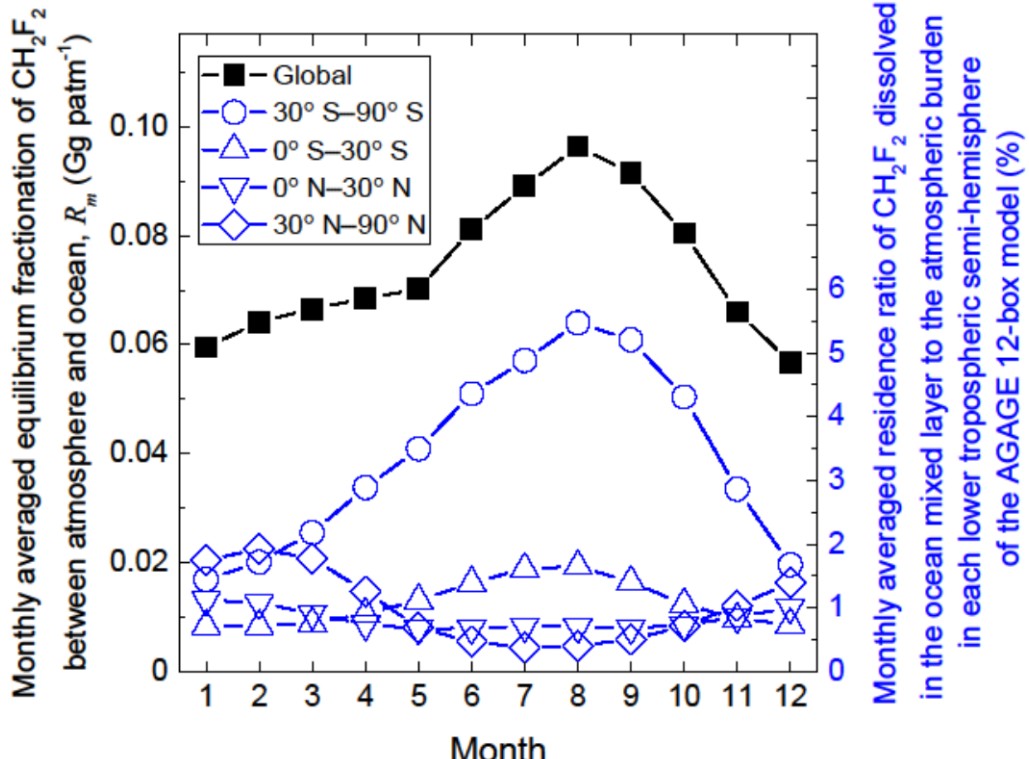

**Figure 6.** Plots of monthly averaged equilibrium fractionation of $CH_2F_2$ between atmosphere and ocean, $R_m$ (Gg patm$^{-1}$) in the global and the hemispheric atmosphere. Right vertical axis represents monthly averaged residence ratio of $CH_2F_2$ dissolved in the ocean mixed layer to the atmospheric burden for each lower tropospheric semi-hemisphere of the AGAGE 12-box model.