# Peer review of "Experimental determination of Henry's law constants of difluoromethane (HFC-32) and the salting-out effects in aqueous salt solutions relevant to seawater"

_Atmospheric Chemistry and Physics, 2017_

## Author Comment (AC1) · 8 Feb 2017

I am sorry because Eqs. (21) and (22) and the corresponding equation in the abstract were mistyped as indicated below.

Eq. (21) should be represented as follows:
$$\ln(K_H(T)/K_{eq}^S(T)) = [0.2261\text{-}0.5176\times(100/T)] \times S^{0.5} + [\text{-}0.0362 + 0.1046\times(100/T)] \times S$$

Eq. (22) should be represented as follows:

$\ln(K_{eq}^S(T)) = -49.7122 + 77.7018 \times (100/T) + 19.1379 \times \ln(T/100) + [-0.2261 + 0.5176 \times (100/T)] \times S^{0.5} + [0.0362 - 0.1046 \times (100/T)] \times S$

In Abstract section, 1st block, lines 6-7, the equation corresponding to Eq. (22) should be represented as follows:
$\ln(K_{eq}^S(T)) = -49.7122 + 77.7018 \times (100/T) + 19.1379 \times \ln(T/100) + [-0.2261 + 0.5176 \times (100/T)] \times S^{0.5} + [0.0362 - 0.1046 \times (100/T)] \times S$

The above corrections are needed because of the mistype of a sign of the right hand of Eq. (21) and the mistype of ($T$/100) instead of the correct term of (100/$T$) for Eq. (21).

Data analysis had been performed by the correct Equations thorough the manuscript; hence, the parameters determined and the results estimated in the manuscript were correct.

I am sorry for my mistake in Eq. (21), Eq. (22) and the Equation corresponding to Eq. (22) in Abstract section.
* * *

---

## Referee Comment (RC1) · Anonymous Referee #1 · 8 Mar 2017

This is a very well-written paper addressing an important topic in atmospheric chemistry. The title and abstract very clearly indicate the purpose and results of the work. The language is clear and concise throughout. The experimental techniques are relatively new but seem quite appropriate to the determination of solubility of gases at very low partial pressures. The techniques are well referenced and described in sufficient detail for others to reproduce the experiments. The main conclusion of the paper seems to be that the solubility of difluoromethane in seawater (accounting for the salting-out effect) is too low to significantly affect the atmospheric burden. The data clearly support this conclusion, although the reviewer is not qualified to comment in detail on the

oceanic modeling referred to as AGAGE.

The reviewer finds fault only in that the author tries to extract more detail from the results than the data justify. Lines 6-7 on page 7 are troubling. data points might be eliminated if some systematic error was found and the measurements repeated, but this does not appear to be the case. A glance at Figure 2 gives the reassuring impression that random errors are quite small for both the IGS and PRV-HS methods. However, there is a small but significant difference in the results at the one temperature where both techniques are used. This shows a systematic error in one or both of the methods. Taking either of the data sets in Figure 2 by itself, a linear fit to ln(K) vs (1/T) would be indicated, but with different slopes, hence different enthalpies of solution. It is only when one uncritically joins the data sets that a curve is seen, which seems to justify the use of the third term in the van't Hoff equation, equation 12. In the fitting equation, equation 13, the number of significant figures reported is much higher than justified for the relatively small number of data points. In nonlinear fitting of this type, most programs report the variance associated with each of the fitting coefficients. if the square-root-of-variance is not small compared to the fitting coefficient, that means that the inclusion of that coefficient is probably not justified. The author does not report these results so it is difficult to determine if the three-term van't Hoff equation is justified. The reviewer believes that a two-term fit would be the highest order justified if each data set were analyzed separately.

The treatment of the salting-out effect is similarly overworked. The data in Figure 4 show the expected qualitative trend with respect to increasing salinity and imply small random errors. The plots in Figure 5 clearly show curvature, since the origin is, in effect, a "free" data point. So far, so good. The generalization of the Sechenov equation (equations 18-22) uses more fitting parameters (four salinity parameters for only five measured salinities!) and more significant figures than are justified. In the reviewer's opinion, lines 9-26, page 9 should be eliminated and the author should simply state that ln(Kh/Keq) varies close to the 0.5 power of salinity, in contrast to the Sechenov

equation. This result is probably new and could be the stimulus for further research.

The paper is almost free of typographical errors. On page 3, line 12, the water quality should be indicated as (resistivity > 18 megohm-cm). On page 6, line 21, "non-linear" is misspelled.

---

## Author Comment (AC2) · 21 Mar 2017

Thank you very much for the constructive comments.

Two main comments are given. Both the comments indicate that the author tries to extract more detail from the results than the data justify. First comment is on the fitting by the three-term van't Hoff equation. Second comment is on the treatment of the salting-out effect. I will replay to each comment as follows.

1. Reply to the comment on the fitting by the three-term van't Hoff equation.

[Figure]

Thank you for the suggestion that the square-root-of-variance of the fitting coefficient should be checked for justifying whether the coefficient should be included in the van't Hoff equation. The square-root-of-variance, that is, standard deviation for each fitting coefficient in equation (12) is as follows:

$\delta a_1 = 5.50; \delta a_2 = 8.32; \delta a_3 = 2.83$

Because the ratio of $\delta a_3/a_3$ is 0.148, the three-term van't Hoff equation is thus justified.

According to the referee's comment, I revise the significant figure of each fitting coefficient in equation (13). I set the least digit of the significant figure to the second decimal place so that the values calculated by equation (13) are consistent with the significant figure of $K_H$. Equation (13) is thus as follows:

$$\ln\left(K_H(T)\right) = -49.71 + 77.70 \times (100/T) + 19.14 \times \ln\left(T/100\right) \quad (13)$$

As the referee indicates, there is, however, a small but significant difference in the results at the one temperature where both techniques are used (around 40 °C). This suggests a systematic error in one or both of the methods. In addition, as the referee suggests, if each data is fitted separately, a two-term fit would reproduce the data.

A two-term fit gives equation (A1) for the IGS method and equation (A2) for the PRV-HS method:

$$\ln\left(K_{\mathsf{H}}(T)\right) = (-10.43 \pm 0.12) + (22.88 \pm 0.35) \times (100/T) \quad \text{(A1)},$$

$$\ln K_{\mathsf{H}}(T) = (-8.88 \pm 0.12) + (18.51 \pm 0.40) \times (100/T) \quad \text{(A2)},$$

where errors of the fitting coefficients represent standard deviation only for non-linear fitting.

Despite these indication and suggestion, I still think that fitting the three-term van't Hoff equation (equation 12) to the data is reasonable because equation (13) is justified as aforementioned and because the following three reasons additionally support it.

First, even if the data in the IGS method is fitted separately, a three-term fit to the data in the IGS method would be justified as equation (A3), although errors of the fitting coefficients are larger than those in equation (13).

$$\ln\left(K_{\mathsf{H}}(T)\right) = (-41.7 \pm 7.2) + (66.8 \pm 10.5) \times (100/T) + (15.1 \pm 3.7) \times \ln\left(T/100\right) \quad \text{(A3)},$$

where errors of the fitting coefficients represent standard deviation only for non-linear fitting. Residual mean square of values of $K_{\mathsf{H}}$ was $1.19 \times 10^{-8}$ for equation (A1) and $5.29 \times 10^{-7}$ for equation (A3).

Second, as seen in Figure 4, if the data for a-seawater at each salinity is fitted separately, the data would be fitted by the three-term van't Hoff equation rather than the two-term van't Hoff equation.

Third, I checked a potential systematic error in the IGS method but did not find it out as follows. Errors of the volumetric flow rate of the gas ($F_{\text{meas}}$) would cause systematic error in the IGS method. But, as described in the experimental section (lines 14-16, on page 4), I had calibrated the soap-bubble meter, which was used to calibrate $F_{\text{meas}}$, by

means of a high-precision film flow meter SF-1U with VP-2U (Horiba, Kyoto, Japan). Other potential systematic error in the IGS method may arise from the redistribution of gaseous $CH_2F_2$ into the test solution (lines 5-14, on page 5). But if the redistribution of gaseous $CH_2F_2$ into the test solution was significant, the value of $K_H(T)$ would have been overestimated because equation (6), not equation (7), was used to deduce the $K_H$ values in this study. Hence, it cannot be the reason for the difference in the values of $K_H$ obtained around 40 °C between in the IGS method and in the PRV-HS method.

Overall, I think that fitting the three-term van't Hoff equation (equation (12)) to the data is reasonable.

2. Reply to the comment that the treatment of the salting-out effect is overworked.

The referee points out the treatment of the salting-out effect is overworked. I agree on this point. The referee has the opinion that lines 9-26, page 9 should be eliminated and the author should simply state that $\ln(K_H/K_{eq})$ varies close to the 0.5 power of salinity. Thank you for the opinion. According to the referee's opinion, almost all the data in Figure 5 is found to be fitted using only one parameter as follows.

According to the referee's opinion, I eliminate equations 18 - 21. Instead, I represent $\ln(K_H/K_{eq})$ by equation A4:

$\ln(K_H(T)/K_{eq}^S(T)) = k_{s1} \times S^{0.5}$ (A4)

where $k_{s1} = a_{s1} \times (100/T)$.

Equation (A4) is used to estimate the amount of $CH_2F_2$ dissolved in the ocean mixed layer in Sect. 3.3. Fitting equation (A4) to all the data in Figure 5 gives $a_{s1} = 0.1343 \pm 0.0013$ (equation (A5)). Errors are standard deviation only for non-linear

fitting.

$$\ln\left(K_{\mathsf{H}}(T)/K_{\mathsf{eq}}^{S}(T)\right) = (0.1343 \pm 0.0013) \times (100/T) \times S^{0.5} \quad \text{(A5)}$$

This fitting reproduces all the data, except for two data at 39.5 °C, in Figure 5, as indicated in Figure 5m. Only one parameter ($a_{\mathsf{s}1}$) is thus found to be required for fitting almost all the data in Figure 5.

Combining equations (13) and (A5) gives equation (A6):

$$\ln\left(K_{\mathsf{eq}}^{S}(T)\right) = -49.71 + (77.70 - 0.134 \times S^{0.5}) \times (100/T) + 19.14 \times \ln\left(T/100\right) \quad \text{(A6)}$$

Based on equation (A6), values of $K_{\mathsf{eq}}^{S}$ and the amount of $CH_2F_2$ dissolved in the ocean mixed layer will be recalculated and the manuscript will be revised.

[Figure]

Fig. 1. (Fig. 5m) Plots of ln(KH(T)/KHS(T)) vs. salinity in a-seawater at each temperature. Bold curves represent the fitting obtained by Eq. (A5)

---

## Author Comment (AC3) · 21 Mar 2017

In my reply (March 21, 2017) to the comment by Referee 1, the sentence relating to equation (A1) is erratum. Please revise it as follows.

(before revision) Residual mean square of values of $K_H$ was $1.19 \times 10^{-8}$ for equation (A1) and $5.29 \times 10^{-7}$ for equation (A3).

(after revision) Residual mean square of values of $K_H$ was $1.19 \times 10^{-6}$ for equation (A1) and $5.29 \times 10^{-7}$ for equation (A3).

---

## Referee Comment (RC2) · Anonymous Referee #2 · 25 Apr 2017

The manuscript by Shuzo Kutsuna entitled "Experimental determination of Henry's Law constants of difluoromethane (HFC-32) and the salting-out effects in aqueous salt solutions relevant to seawater" describes nice laboratory measurements of Henry's Law, and modified Setchenov constants as a function of temperature and salinity. The experiments are conducted for a single (constant) mole ratio composed of NaCl, MgCl2, Na2SO4, CaCl2 and KCl, resembling artificial seawater. The total salt concentration of this mixture is varied to resemble salinity conditions that include salt concentrations found in the ocean (up to 51.3 permille). The range of temperatures probed (276K-353K) provides good signal-to-noise to estimate the enthalpy for dissolution of CH2F2

into water. The manuscript argues convincingly, that a single Setchenov salting constant does not describe the salting out effect over the full range of conditions probed. Instead, a parameterization based on a 7-variable global fit is chosen to represent the equilibrium partitioning as a function of salinity, and temperature. This manuscript is suitable for publication in ACP, and I recommend publication after the following comments have been addressed. Comments:

1) The overall quality of the measurements seems high, but the manuscript would benefit from an explicit discussion of experimental error. At present, there is essentially no discussion of experimental error. What are the parameters that limit the accuracy of the inert-gas stripping (IGS) method? Of the stripping column apparatus? And of the phase ratio variation headspace method (PRV-HS) method? Do error bars reflect statistics only, or also potential sources of systematic bias?

2) The data seems to be precise, but is not necessarily accurate. For example, there does appear to be a small – yet significant – offset between PRV-HS and IGS method in Figure 2 at 312K, where there is overlap between both methods. This offset seems to be significant based on the error bars shown. Is IGS believed to be more accurate, why?

3) Does the fit according to (Eq13) take into account the relative weight of error bars?

4) What is the reason for the large variation in the size of error bars in Fig. 5?

5) The S0.5 components of the fit (deviation from Setchenov) is strongest at warm temperatures, and smallest at low temperatures. This is an interesting observation, that warrants discussion. Beyond the empirical finding there is virtually no discussion of these deviations. What are possible causes? What is its relevance?

6) The discussion in Sect. 3.3 assumes solubility equilibrium with the atmosphere over the full depth of the ocean mixed layer. How deep is this mixed ocean layer in the model? Concentration gradients in DOC are visible in field data already at a few ten

meters from the ocean surface. Does this mean the model estimates an upper limit?

7) The conclusion that 5% of the atmospheric burden of CH2F2 would reside in the ocean mixed layer in the southern semi-hemispheric lower troposphere during winter seems to be an upper limit, and should be worded as such. How much lower could this upper limit be?

8) For a gas with an atmospheric lifetime of 5.4 years, that is mostly emitted in the Northern Hemisphere it seems surprising that the dissolution of CH2F2 into the ocean should affect estimates of CH2F2 emissions in the Southern Hemisphere and their seasonal variability, because the atmospheric concentrations that reach the Southern Hemisphere are also affected by transport, and chemical removal, and related uncertainties. This should be mentioned.

Specific comments:

P2, L28: 'first' is written twice

P5, L8: add errors for numbers. See comments #1, #2 and add typical values, their units, and uncertainties of variables for the key equations throughout the manuscript.

P7, L6: What statistical test for outliers was applied? Six or fewer points were removed at each temperature. Out of how many? Please add.

Eq(17). For sake of discussion, can a kS value be given here? And what is the effect of including kS vs kS1, kS2 in the model – does it make a difference?

—————————————————————

---

## Author Comment (AC4) · 9 May 2017

Thank you very much for the constructive comments.

The comments are on description of errors and potential systematic bias for the determined values, discussion of the salting-out behavior observed, estimate of the amount of $CH_2F_2$ dissolved in the ocean mixed layer, and relation between dissolution of $CH_2F_2$ in the ocean and estimation of $CH_2F_2$ emissions in the Southern Hemisphere and its seasonal variability.

I will reply to each comment as follows. Parts of texts and numbers revised in the manuscript are marked in blue.

**R1. Reply to the comment on discussion of experimental error. What are the parameters that limit the accuracy of the inert-gas stripping (IGS) method? Of the stripping column apparatus? And of the phase ratio variation headspace (PRV-HS) method? Do error bars reflect statistics only, or also potential sources of systematic bias?**

Thank you for the comments.

I reply to the comments in the sequence: (1) on the parameters that limit the accuracy of the IGS method; (2) on the parameters that limit the accuracy of the PRV-HS method; and (3) on error bars.

(1) the parameters that limit the accuracy of the IGS methods

The parameters that limit the accuracy of the IGS methods are temperature of the test solution ($T$) and flow rate of purge gas ($F$).

The accuracy of $T$ ($\delta T$) are within 0.2 K and may give potential systematic bias of ±0.5 to ±0.6 % ($\delta K_{eq}/K_{eq}$), where $\delta K_{eq}$ indicates an error of $K_{eq}$.

For $F$, the accuracy of $F_{meas}$ is estimated to be within 1% from the accuracy of the high-precision film flow meter SF-1U with VP-2U used for calibrating the soap flow meter. Errors in the term of $\frac{P_{meas}-h_{meas}}{P_{hs}-h} \times \frac{T}{T_{meas}}$ in Eq (3) are estimated at ca. ±1 %. Hence, the accuracy of $F$ ($\delta F$) are estimated to be within 1.4 % and may give potential systematic bias of ±1.4 % of $\delta K_{eq}/K_{eq}$.

Values of $\delta K_{eq}/K_{eq}$ due to both $\delta T$ and $\delta F$ may thus have potential systematic bias of ca. ±2%.

(2) the parameters that limit the accuracy of the PRV-HS methods

The parameters that limit the accuracy of the PRV-HS methods are temperature of the test solution ($T$) and volume of the vials used ($V$).

Although the apparatus used (Agilent, HP7694) was expected to keep $T$ constant, the accuracy of $T$ may not be certified. I have applied the same apparatus to determination of the $K_H$ values for some HCFCs such as HCFC-123 using the PRV-HS methods [Kutsuna, S. *Int. J. Chem. Kinet.*, 45, 440-451, 2013]. On the basis of the $K_H$ values thus determined and comparison between them and the reported values for HCFC-123, errors of $T$ are estimated to be within ca. 2 K. These errors of $T$ may give potential systematic bias of ca. ±4 % ($\delta K_{eq}/K_{eq}$) at 313 K and ca. ±3 % ($\delta K_{eq}/K_{eq}$) at 353 K.

Errors for $V$ ($\delta V$) are estimated to be less than 1 %, and these errors may give potential systematic bias of less than 1 % of $\delta K_{eq}/K_{eq}$.

Accordingly, for the PRV-HS methods, values of $\delta K_{eq}/K_{eq}$ due to both $\delta T$ and $\delta V$ may have potential systematic bias of ca. ±4%.

(3) Error bars in Figure 2

Error bars in Figure 2 reflect statics only (Error_S) in the original manuscript. Error bars in Figure 2m represent errors (Error_T) reflecting both Error_S and potential systematic bias (Error_B). Values of Error_T are also indicated in Tables 1m, S1m and 3m. Values of Error_T are calculated by (Error_S + Error_B) rather than $\sqrt{(Error\_S)^2 + (Error\_B)^2}$ because

Error_B is potential systematic bias.

**Table 1m.** The average of values of $F/(k_1RTV)$ obtained for $V$ value of 0.350 dm³ and 0.300 dm³ and the $K_H(T)$ value derived from Eq. (13) at each temperature. $N$ represents number of experimental runs for the average.

| $T$ (K) | $F / (k_1RTV)$ | | | | $K_H(T)$ (M atm$^{-1}$) |
|---|---|---|---|---|---|
| | $V = 0.350$ | | $V = 0.300$ | | |
| | average [a, b] | $N$ [c] | average [a] | $N$ [c] | From Eq. (13) [d] |
| 276.15 | 0.119 ± 0.006 (0.008) | 21 (2) | 0.117 ± 0.006 (0.008) | 11 (0) | 0.118 ± 0.003 |
| 278.35 | 0.107 ± 0.005 (0.007) | 18 (3) | 0.110 ± 0.005 (0.007) | 14 (0) | 0.108 ± 0.002 |
| 283.65 | 0.093 ± 0.003 (0.005) | 27 (5) | 0.092 ± 0.001 (0.003) | 5 (0) | 0.094 ± 0.002 |
| 288.65 | 0.082 ± 0.006 (0.008) | 41 (5) | 0.084 ± 0.006 (0.008) | 12 (0) | 0.082 ± 0.002 |
| 293.45 | 0.071 ± 0.001 (0.002) | 15 (8) | 0.071 ± 0.001 (0.002) | 5 (0) | 0.072 ± 0.002 |
| 298.15 | 0.064 ± 0.002 (0.003) | 30 (6) | 0.067 ± 0.005 (0.006) | 12 (0) | 0.064 ± 0.002 |
| 303.05 | 0.057 ± 0.003 (0.004) | 16 (0) | 0.056 ± 0.005 (0.006) | 4 (0) | 0.058 ± 0.002 |
| 307.95 | 0.051 ± 0.001 (0.002) | 12 (6) | 0.054 ± 0.004 (0.005) | 10 (0) | 0.052 ± 0.002 |
| 312.65 | 0.046 ± 0.001 (0.002) | 13 (3) | 0.047 ± 0.001 (0.002) | 4 (0) | 0.048 ± 0.001 |

a. Errors are 2σ for the average only.; b. Number in parenthesis represents an error reflecting both 2σ for the average and potential systematic bias.; c. Number in parenthesis represents number of experimental runs excluded for the average.; d. Errors are 95% confidence level for the regression only.

**Table S1m.** $L_i$ values for various $V_i/V_0$ ratios at various temperatures, slopes and intercepts for linear regression with respect to Eq. (10), $K_H(T)$ values calculated from the slopes and intercepts, and $K_H(T)$ values and the errors at 95% confidence level estimated by non-linear fitting the two datasets simultaneously at each temperature (Fig. S4) with respect to Eq. (11).

| $T$ (K) | $L_i$ (a.u.) [*] | | | | | | Eq. (10) Intercept | Eq. (10) Slope | $K_H$ (M atm$^{-1}$) | | |
|---|---|---|---|---|---|---|---|---|---|---|---|
| | $V_i/V = 0.421$ | 0.351 | 0.280 | 0.210 | 0.140 | 0.070 | | | Eq. (10) | Eq. (11) [**, ***] | Eq. (13) [**] |
| 353 | 3.226±0.002 | 3.270±0.026 | 3.330±0.004 | 3.391±0.008 | 3.462±0.014 | 3.526±0.009 | 3.581 | –0.870 | 0.026 | 0.027 ±0.002 (±0.003) | 0.031 ±0.003 |
| | 2.044±0.006 | 2.050±0.012 | 2.112±0.010 | 2.132±0.009 | 2.186±0.021 | 2.209±0.011 | 2.248 | –0.513 | 0.027 | | |
| 343 | 3.000±0.018 | 3.025±0.009 | 3.070±0.008 | 3.089±0.015 | 3.117±0.015 | 3.148±0.018 | 3.179 | –0.423 | 0.031 | 0.031 ±0.001 (±0.002) | 0.033 ±0.002 |
| | 1.949±0.004 | 1.955±0.005 | 1.968±0.003 | 1.998±0.004 | 2.020±0.002 | 2.030±0.009 | 2.050 | –0.258 | 0.031 | | |
| 333 | 3.247±0.018 | 3.234±0.018 | 3.243±0.015 | 3.241±0.010 | 3.247±0.009 | 3.223±0.013 | 3.231 | 0.034 | 0.037 | 0.036 ±0.003 (±0.004) | 0.037 ±0.002 |
| | 3.080±0.009 | 3.044±0.006 | 3.082±0.005 | 3.127±0.009 | 3.113±0.008 | 3.134±0.014 | 3.149 | –0.213 | 0.034 | | |
| 323 | 3.208±0.011 | 3.190±0.008 | 3.133±0.010 | 3.134±0.011 | 3.092±0.008 | 3.093±0.006 | 3.055 | 0.355 | 0.042 | 0.043 ±0.002 (±0.004) | 0.042 ±0.001 |
| | 3.357±0.010 | 3.289±0.014 | 3.275±0.005 | 3.233±0.004 | 3.226±0.016 | 3.160±0.001 | 3.135 | 0.496 | 0.044 | | |
| 313 | 3.245±0.018 | 3.185±0.013 | 3.100±0.015 | 3.022±0.012 | 2.995±0.012 | 2.915±0.011 | 2.848 | 0.935 | 0.052 | 0.052 ±0.003 (±0.005) | 0.049 ±0.001 |
| | 2.162±0.031 | 2.134±0.010 | 2.060±0.014 | 2.029±0.018 | 1.992±0.010 | 1.925±0.018 | 1.896 | 0.612 | 0.052 | | |

[*] Errors are 2σ for the regression only.; ** Errors are those at 95% confidence level for the regression only.; *** Number in parenthesis represents both errors at 95% confidence level for the regression and potential systematic bias.

[Figure]

**Figure 2m.** van't Hoff plot of the $K_H$ values obtained by the IGS method and the PRV-HS method. Bold curve displays the fitting of the data obtained by the IGS method and the PRV-HS method (Eq. (13)). Dashed curves display upper and lower 95% confidence limit of the above fitting by Eq. (12). Error bars of the data by the IGS method represent both 2σ for the average and potential systematic bias. Error bars of the data by PRV-HS method represent both errors at 95% confidence level for the regression and potential systematic bias.

**R2. Reply to the comment 2 that there does appear to be a small – yet significant – offset between PRV-HS and IGS method in Figure 2 at 312 K and why IGS is believed to be more accurate.**

Thank you for the comments.

There appears to be a small - yet significant - offset between PRV-HS and IGS method at 312 K. This point is also commented by Referee 1. For the PRV-HS methods, values of $\delta K_{eq}/K_{eq}$ may have potential systematic bias of ca. ±4%, which results mostly from the accuracy of temperature of the test solution, as aforementioned (R1. (2)). For the IGS method, values of $\delta K_{eq}/K_{eq}$ may have potential systematic bias of ca. ±2%. The IGS method is thus believed to be more accurate. Potential systematic bias in both the PRV-HS method and the IGS method could be a reason why there is the small offset between PRV-HS and IGS method at 312 K.

**R3. Reply to the comment 3 on the fit according to Eq. (13)**

Thank you for the comment.

The fit according to Eq. (13) does not take into account the relative weight of error bars.

**R4. Reply to the comment 4 on the reason for the large variation in the size of error bars in Fig. 5.**

Thank you for the comment.

As Referee 2 comments, there are the large variation in the size of error bars in Fig. 5. Ratio among error bars of the

data at the same temperature is up to the maximum value of 4.5: error bars are 0.084 for 8.921‰ and 0.019 for 51.534‰ at 10.5 °C. Error bars for the data at 8.921‰ tend to be large and error bars for the data at 51.534‰ tend to be small: this reflects statics errors of the data at 8.921‰ and 51.534‰.

Errors of the data in Fig. 5 represents statics only (Error_S). As replied in R1, errors from both statics (Error_S) and potential systematic bias of ±2% (Error_B) will be used as errors (Error_T) for the data in Fig. 5: (Error_T) = (Error_S) + (Error_B). Error bars of the data in Fig. 5m represent Error_T. As seen in Fig. 5m, the ratios among error bars of the data at the same temperature are smaller than the corresponding ratios in Fig. 5. For example, the ratio of error bars between at 8.921‰ and 51.534‰ at 10.5 °C is 2.7 while it is 4.5 in Fig. 5 as aforementioned.

In the revised manuscript, error bars will be represented by Error_T in Fig. 5m.

[Figure]

**Figure 5m.** Plots of $\ln(K_H(T)/K_H^S(T))$ vs. salinity in a-seawater at each temperature. Bold curves represent the fitting obtained by Eq. (22). Error bars represent errors reflecting both 2σ for the average and potential systematic bias of $K_{eq}^S$.

**R5. Reply to the comment 5 on the discussion on the $S^{0.5}$ components of the fit (deviation from Setchenow).**

Thank you for the comment.

The reason why $\ln(K_H/K_{eq}^S)$ is proportional to $S^{0.5}$ rather than $S$ is still unclear.

I will describe a potential reason for this proportionality simply in the text, and make discussion in *Supporting Information* as follows.

L20-21, P9:

The reason for this salting-out effect of $CH_2F_2$ solubility in a-seawater is not clear. Specific properties of $CH_2F_2$ −small molecular volume, which results in small work of cavity creation (Graziano, 2004; 2008), and large solute-solvent attractive potential energy in water and a-seawater− may cause deviation from Setchenow relationship (*Supporting Information*).

In *Supporting Information*:

I will calculate Ben-Naim standard Gibbs energy $\Delta G^{\bullet}$, enthalpy $\Delta H^{\bullet}$, and entropy $\Delta S^{\bullet}$ changes for dissolution of $CH_2F_2$ in water because these values correspond to the values for the transfer from a fixed position in the gas phase to a fixed position in water. Values of $\Delta G^{\bullet}$, $\Delta H^{\bullet}$, and $\Delta S^{\bullet}$ are calculated on the basis of the Ostwald solubility coefficient, $L(T)$, as follows.

$$\ln(L(T)) = \ln\left(RTK_{eq}{}^{S}(T)\right) \tag{B1}$$

$$\Delta G^{\bullet} = R'T\ln(L(T)) \tag{B2}$$

$$\Delta H^{\bullet} = -\frac{\partial}{\partial(^1/_T)}\left(\frac{\Delta G^{\bullet}}{T}\right) \tag{B3}$$

$$\Delta S^{\bullet} = \frac{\Delta H^{\bullet} - \Delta G^{\bullet}}{T} \tag{B4}$$

where both $R$ and $R'$ represent gas constant but their units are different: $R = 0.0821$ in atm dm$^3$ K$^{-1}$ mol$^{-1}$; $R' = 8.314$ in J K$^{-1}$ mol$^{-1}$.

Combining Eqs. (B1), (B2), (B3), and (B4) with Eqs. (14) and (15), $\Delta G^{\bullet}$ (kJ mol$^{-1}$), $\Delta H^{\bullet}$ (kJ mol$^{-1}$), and $\Delta S^{\bullet}$ (J mol$^{-1}$ K$^{-1}$) are represented by $\Delta G_{sol}$ and $\Delta H_{sol}$ as follows:

$$\Delta G^{\bullet} = \Delta G_{sol} + R'T\ln(RT) \tag{B5}$$

$$\Delta H^{\bullet} = \Delta H_{sol} + R'T \tag{B6}$$

$$\Delta S^{\bullet} = \frac{\Delta H_{sol} - \Delta G_{sol}}{T} + R'T - R'T\ln(RT) \tag{B7}$$

Values of $\Delta G^{\bullet}$, $\Delta H^{\bullet}$, and $\Delta S^{\bullet}$ calculated at 298 K are listed in Table S2. Table S2 also lists values of $\Delta G^{\bullet}$, $\Delta H^{\bullet}$, and $\Delta S^{\bullet}$ reported for $CH_3F$ and $C_2H_6$ (Graziano, 2004) and $CH_4$ (Graziano, 2008) at 298 K. The chemicals, which having a methyl group, in Table 2 are classified into two groups ($CH_2F_2$ and $CH_3F$; $CH_4$ and $C_2H_6$) according to $\Delta G^{\bullet}$.

**Table S2. Ben-Naim standard hydration Gibbs energy $\Delta G^{\bullet}$, enthalpy $\Delta H^{\bullet}$, and entropy $\Delta S^{\bullet}$ changes for dissolution of $CH_2F_2$ at 298 K determined here and the corresponding values reported for $CH_3F$ and $C_2H_6$ (Granziano, 2004) and $CH_4$ (Graziano, 2008).**

| | $\Delta G^{\bullet}$ (kJ mol$^{-1}$) | $\Delta H^{\bullet}$ (kJ mol$^{-1}$) | $\Delta S^{\bullet}$ (J K$^{-1}$ mol$^{-1}$) | $\Delta G_c$ (kJ mol$^{-1}$) | $E_a$ (kJ mol$^{-1}$) | $\Delta H^h$ (kJ mol$^{-1}$) |
|---|---|---|---|---|---|---|
| $CH_2F_2$ | −1.1 | −14.7 | −45.4 | | | |
| $CH_3F$ | −0.9 | −15.8 | −50.0 | 23.3 | −24.3 | 8.5 |
| $CH_4$ | 8.4 | −10.9 | −64.7 | 22.9 | −14.5 | 3.7 |
| $C_2H_6$ | 7.7 | −17.5 | −84.5 | 28.4 | −20.7 | 3.2 |

Table 2 lists values of $\Delta G_c$, $E_a$ and $\Delta H^h$ deduced using a scaled particle theory (Granziano, 2004; 2008). $\Delta G_c$ is the work of cavity creation to insert a solute in a solvent. $E_a$ is a solute-solvent attractive potential energy and accounts for the solute-solvent interactions consisting of dispersion, dipole-induced dipole, and dipole-dipole contributions. $\Delta H^h$ is enthalpy of solvent molecules reorganization caused by solute insertion. The solvent reorganization mainly involves a rearrangement of H-bonds.

$\Delta G_c$ is entropic in nature in all liquids, being a measure of the excluded volume effect due to a reduction in the spatial configurations accessible to liquid molecules upon cavity creation. Hence, $C_2H_6$ has larger value of $\Delta G_c$ than $CH_3F$ and $CH_4$. $\Delta G_c$, $E_a$, and $\Delta H^h$ are related to $\Delta G^{\bullet}$ and $\Delta H^{\bullet}$ as follows (Graziano, 2008):

$$\Delta G^{\bullet} = \Delta G_c + E_a \tag{B8}$$

$$\Delta H^{\bullet} = E_a + \Delta H^h \tag{B9}$$

Table S10m thus suggests that smaller value of $\Delta G^{\bullet}$ of $CH_3F$ than $CH_4$ is due to large solute-solvent attractive potential energy ($-E_a$) of $CH_3F$.

Graziano (2008) definitively explained the salting-out of $CH_4$ by sodium chloride at molecular level on the basis of a

scaled particle theory. He explained that $\Delta G^\bullet$ increase was linearly related to the increase in the volume packing density of the solutions ($\xi_3$) with adding NaCl. Such a increase of $\Delta G^\bullet$ is probably the case for salting-out of $CH_2F_2$ by a-seawater observed in this study. He also explained that $E_a$ was linearly related to the increase in $\xi_3$ assuming that a fraction of the dipole-induced dipole attractions could be taken into account by the parameterization of the dispersion contribution.

I think the possibility that $E_a$ may be nonlinearly related to the increase in $\xi_3$ because of dipole-dipole interaction between $CH_2F_2$ and solvents. Temperature dependence in Eq. (23) suggests that salting-out effect of $CH_2F_2$ by a-seawater is enthalpic. Eqs. (23) and (B9) thus suggests that the salting-out of $CH_2F_2$ is mostly related to change in $E_a$. $CH_2F_2$ has relatively small value of $\Delta G_c$ because of its small molecular volume compared to other chemicals such as $C_2H_6$. Accordingly, $\Delta G^\bullet$, that is, solubility of $CH_2F_2$ would depend on $E_a$ rather than $\Delta G_c$. Therefore, I think that specific properties of $CH_2F_2$ –small molecular volume, which results in small work of cavity creation (Graziano, 2004; 2008), and large solute-solvent attractive potential energy in water and a-seawater– may cause deviation from Setchenow relationship.

The following two references will be cited both in the manuscript and in *Supporting Information*.

Graziano, G.: Case study of enthalpy–entropy noncompensation. Journal of Chemical Physics, 120, 4467-4471, doi: 10.1063/1.1644094, 2004.

Graziano, G.: Salting out of methane by sodium chloride: A scaled particle theory study. Journal of Chemical Physics, 129, 084506, doi: 10.1063/1.2972979, 2008.

**R6. Reply to the comments 6: how deep is this mixed ocean layer in the model? Does this mean the model estimates an upper limit?**

Thank you for the comments.

The depth of the ocean mix layer in the model is 10 to 600 m. The depth distribution of $CH_2F_2$ dissolved in the ocean mixed layer in each semi-hemisphere is listed in Tables S3 (30° S–90° S), S4 (30° S–0° S), S5 (0° N–30° S) and S6 (30° N–90° N). As seen in these tables, the $CH_2F_2$ dissolved in the ocean mixed layer resides mostly in less than 300 m depth. For example, for the southern semi-hemisphere (30° S–90° S) (Table S3), in August, when the amount of $CH_2F_2$ dissolved in the ocean mixed layer is maximum, 66% of the $CH_2F_2$ dissolved in the mixed layer would reside between 200 m and 300 m depth, and 91 % of the $CH_2F_2$ dissolved in the ocean mixed layer is expected to reside in less than 300 m depth.

As Referee 2 pointed out, model estimates mean an upper limit of the amount of $CH_2F_2$ dissolved in the ocean mixed layer. This point will be clearly described in the revised manuscript as follows.

P10, L23:

As seen in Figure 6, in the southern semi-hemispheric lower troposphere (30° S−90° S), at least 5 % of the atmospheric burden of $CH_2F_2$ would reside in the ocean mixed layer in the winter, and the annual variance of the $CH_2F_2$ residence ratio would be 4%. These ratios are, in fact, upper limits because $CH_2F_2$ in the ocean mixed layer may be undersaturated. It takes days to a few weeks after a change in temperature or salinity for oceanic surface mixed layers to come to equilibrium with the present atmosphere, and equilibration time increases with depth of the surface mixed layer (Fine, 2011). In the estimation using the gridded data here, >90 % of $CH_2F_2$ in the ocean mixed layer would reside in less than 300 m depth (Tables S3, S4, S5 and S6).

Haine and Richards (1995) demonstrated that seasonal variation in ocean mixed layer depth was the key process which affected undersaturation and supersaturation of chlorofluorocarbon 11 (CFC-11), CFC-12 and CFC-113 by use of a one-dimensional slab mixed model. As described above, >90 % of $CH_2F_2$ in the ocean mixed layer is expected to reside in less than 300 m depth. According to the model calculation results by Haine and Richards (1995), saturation of $CH_2F_2$ would

be >0.9 for the ocean mixed layer with less than 300 m depth. The saturation of $CH_2F_2$ in the ocean mixed layer is thus estimated to be at least 0.8. In the southern semi-hemispheric lower troposphere (30° S–90° S), therefore, at least 4 % of the atmospheric burden of $CH_2F_2$ would reside in the ocean mixed layer in the winter, and the annual variance of the $CH_2F_2$ residence ratio would be 3%.

The following two references will be cited in the manuscript.

Fine, R. A.: Observations of CFCs and $SF_6$ as ocean tracers. Annual Review of Marine Science, 3, 173-195, doi:10.1146/annurev.marine.010908.163933, 2011.

Haine, T. W. N. and Richards, K. J.: The influence of the seasonal mixed layer on oceanic uptake of CFCs. Journal of Geophysical Research, 100, 10727-10744, doi:10.1029/95JC00629, 1995.

**Table S3. Monthly amount of $CH_2F_2$ dissolved in the ocean mixed layer at solubility equilibrium with the atmospheric $CH_2F_2$ (1 patm) and the depth distribution of the $CH_2F_2$ dissolved in the southern semi-hemisphere (90°S - 30°S).**

| | Amount (Gg patm$^{-1}$) | Depth distribution of $CH_2F_2$ dissolved in the ocean mixed layer (%) | | | | | |
|---|---|---|---|---|---|---|---|
| | | 10 - 100 m | 100 - 200 m | 200 - 300 m | 300 - 400 m | 400 - 500 m | 500 - 600 m |
| January | 0.0169 | 94.9 | 2.9 | 1.0 | 0.5 | 0.3 | 0.3 |
| February | 0.0201 | 92.1 | 3.6 | 2.9 | 1.0 | 0.3 | 0.0 |
| March | 0.0255 | 87.8 | 9.2 | 1.7 | 0.7 | 0.2 | 0.4 |
| April | 0.0338 | 66.5 | 31.8 | 1.1 | 0.2 | 0.1 | 0.2 |
| May | 0.0409 | 48.5 | 48.1 | 2.2 | 0.8 | 0.3 | 0.0 |
| June | 0.0510 | 26.8 | 62.7 | 8.0 | 1.7 | 0.8 | 0.1 |
| July | 0.0571 | 14.1 | 69.3 | 12.2 | 3.3 | 0.9 | 0.1 |
| August | 0.0640 | 8.5 | 65.8 | 17.0 | 6.2 | 2.3 | 0.2 |
| September | 0.0609 | 13.5 | 61.0 | 14.6 | 8.2 | 2.7 | 0.0 |
| October | 0.0504 | 24.7 | 58.6 | 12.1 | 2.9 | 1.4 | 0.3 |
| November | 0.0335 | 60.4 | 30.5 | 4.6 | 2.2 | 2.3 | 0.1 |
| December | 0.0196 | 95.1 | 4.3 | 0.4 | 0.2 | 0.0 | 0.0 |

**Table S4. Monthly amount of $CH_2F_2$ dissolved in the ocean mixed layer at solubility equilibrium with the atmospheric $CH_2F_2$ (1 patm) and the depth distribution of the $CH_2F_2$ dissolved in the southern semi-hemisphere (30°S - 0°S).**

| | Amount (Gg patm$^{-1}$) | Depth distribution of $CH_2F_2$ dissolved in the ocean mixed layer (%) | | | | | |
|---|---|---|---|---|---|---|---|
| | | 10 - 100 m | 100 - 200 m | 200 - 300 m | 300 - 400 m | 400 - 500 m | 500 - 600 m |
| January | 0.0084 | 99.6 | 0.4 | 0 | 0 | 0 | 0 |
| February | 0.0084 | 99.7 | 0.3 | 0 | 0 | 0 | 0 |
| March | 0.0089 | 100.0 | 0 | 0 | 0 | 0 | 0 |
| April | 0.0106 | 100.0 | 0 | 0 | 0 | 0 | 0 |
| May | 0.0131 | 100.0 | 0 | 0 | 0 | 0 | 0 |
| June | 0.0163 | 97.1 | 2.9 | 0 | 0 | 0 | 0 |
| July | 0.0189 | 80.1 | 19.9 | 0 | 0 | 0 | 0 |
| August | 0.0193 | 73.1 | 26.9 | 0 | 0 | 0 | 0 |
| September | 0.0165 | 82.2 | 17.8 | 0 | 0 | 0 | 0 |
| October | 0.0124 | 94.6 | 5.4 | 0 | 0 | 0 | 0 |
| November | 0.0097 | 99.9 | 0.1 | 0 | 0 | 0 | 0 |
| December | 0.0087 | 100.0 | 0 | 0 | 0 | 0 | 0 |

**Table S5. Monthly amount of CH$_2$F$_2$ dissolved in the ocean mixed layer at solubility equilibrium with the atmospheric CH$_2$F$_2$ (1 patm) and the depth distribution of the CH$_2$F$_2$ dissolved in the southern semi-hemisphere (0°N - 30°N).**

| | Amount (Gg patm$^{-1}$) | Depth distribution of CH$_2$F$_2$ dissolved in the ocean mixed layer (%) | | | | | |
|---|---|---|---|---|---|---|---|
| | | 10 - 100 m | 100 - 200 m | 200 - 300 m | 300 - 400 m | 400 - 500 m | 500 - 600 m |
| January | 0.0132 | 96.4 | 3.6 | 0 | 0 | 0 | 0 |
| February | 0.0126 | 95.9 | 4.1 | 0 | 0 | 0 | 0 |
| March | 0.0107 | 98.7 | 1.3 | 0 | 0 | 0 | 0 |
| April | 0.0087 | 99.8 | 0.2 | 0 | 0 | 0 | 0 |
| May | 0.0079 | 100.0 | 0 | 0 | 0 | 0 | 0 |
| June | 0.0080 | 100.0 | 0 | 0 | 0 | 0 | 0 |
| July | 0.0084 | 100.0 | 0 | 0 | 0 | 0 | 0 |
| August | 0.0082 | 100.0 | 0 | 0 | 0 | 0 | 0 |
| September | 0.0080 | 100.0 | 0 | 0 | 0 | 0 | 0 |
| October | 0.0086 | 100.0 | 0 | 0 | 0 | 0 | 0 |
| November | 0.0100 | 100.0 | 0 | 0 | 0 | 0 | 0 |
| December | 0.0118 | 100.0 | 0 | 0 | 0 | 0 | 0 |

**Table S6. Monthly amount of CH$_2$F$_2$ dissolved in the ocean mixed layer at solubility equilibrium with the atmospheric CH$_2$F$_2$ (1 patm) and the depth distribution of the CH$_2$F$_2$ dissolved in the southern semi-hemisphere (30°N - 90°N).**

| | Amount (Gg patm$^{-1}$) | Depth distribution of CH$_2$F$_2$ dissolved in the ocean mixed layer (%) | | | | | |
|---|---|---|---|---|---|---|---|
| | | 10 - 100 m | 100 - 200 m | 200 - 300 m | 300 - 400 m | 400 - 500 m | 500 - 600 m |
| January | 0.0205 | 41.3 | 50.1 | 7.0 | 1.4 | 0.2 | 0.0 |
| February | 0.0225 | 34.5 | 55.3 | 7.1 | 2.3 | 0.6 | 0.2 |
| March | 0.0208 | 49.7 | 42.3 | 4.9 | 1.7 | 0.7 | 0.6 |
| April | 0.0147 | 79.7 | 17.6 | 1.7 | 0.4 | 0.0 | 0.6 |
| May | 0.0081 | 90.1 | 9.9 | 0 | 0 | 0 | 0 |
| June | 0.0055 | 97.7 | 2.3 | 0 | 0 | 0 | 0 |
| July | 0.0045 | 96.6 | 3.4 | 0 | 0 | 0 | 0 |
| August | 0.0048 | 94.4 | 5.6 | 0 | 0 | 0 | 0 |
| September | 0.0059 | 97.7 | 2.3 | 0 | 0 | 0 | 0 |
| October | 0.0084 | 99.6 | 0.4 | 0 | 0 | 0 | 0 |
| November | 0.0121 | 89.6 | 10.4 | 0.1 | 0 | 0 | 0 |
| December | 0.0163 | 71.0 | 26.1 | 2.9 | 0 | 0 | 0 |

**R7. Reply to the comment 7 on how much departure from saturation equilibrium the oceanic mixed layer in the model is.**

Thank you for the comment.

As described in R6, it takes days to a few weeks after a change in temperature or salinity for oceanic surface mixed layers to come to equilibrium with the present atmosphere, and equilibration time increases with depth of the surface mixed layer (Fine, 2011).

Haine and Richards (1995) demonstrated that the seasonal variation in ocean mixed layer depth was the key process which affected undersaturation and supersaturation of chlorofluorocarbon 11 (CFC-11), CFC-12 and CFC-113 by use of a one-dimensional slab mixed model. Specifically, the mixed layer deepening in autumn would cause undersaturation in the mixed layer. In the estimation, >90 % of CH$_2$F$_2$ in the ocean mixed layer is expected to reside in less than 300 m depth (Tables S3, S4, S5 and S6). According to the report by Haine and Richards (1995), saturation of CH$_2$F$_2$ would be >0.9 for the ocean mixed layer with less than 300 m depth. The saturation of CH$_2$F$_2$ in the ocean mixed layer is thus estimated to be at least 0.8.

The manuscript will be revised, as described in R6, and Fine (2011) and Haine and Richards (1995) will be cited.

**R8. Replay to the comment 8 on how the dissolution of $CH_2F_2$ into the ocean should affect estimation of $CH_2F_2$ emissions in the Southern Hemisphere and their seasonal variability.**

Thank you for the comment.

As Referee 2 pointed out, the atmospheric concentrations that reach the Southern Hemisphere are also affected by transport, chemical removal, and related uncertainties; this should be mentioned.

I will first describe how the dissolution of $CH_2F_2$ into the ocean may affect estimation of $CH_2F_2$ emissions in the Southern Hemisphere and their seasonal variability, and then I will show the revised text.

In 2012, atmospheric concentrations of $CH_2F_2$ in the Northern Hemisphere are by >30% higher than in the Southern Hemisphere (O'Doherty et al., 2014); the strong inter-hemisphere gradient indicates that emissions of $CH_2F_2$ are predominantly in the Northern Hemisphere. In the AGAGE 12 box model (Rigby et al., 2013), transport of $CH_2F_2$ is dominated by eddy diffusion between the boxes in the model. The seasonal eddy diffusion parameters between the Northern Hemisphere and the Southern Hemisphere in the model are 187 to 568 days in lower troposphere, and 81 to 109 days in upper troposphere (Rigby et al., 2013).

The rate of increase in atmospheric concentration of $CH_2F_2$ due to the emission of $CH_2F_2$ in the Southern Hemisphere, which is denoted as $RE_{south}$ hereafter, is thus more sensitive to change in atmospheric concentrations of $CH_2F_2$ in the Southern Hemisphere than those in the Northern Hemisphere, partly because $CH_2F_2$ is removed through gas phase reactions with OH (partial atmospheric lifetime of 5.5 years). Furthermore, $RE_{south}$ would range small values such as a few % $y^{-1}$ or less because emissions of $CH_2F_2$ are predominantly in the Northern Hemisphere and because, in 2012, the rate of increase in the global mean mole fraction of $CH_2F_2$ was 17% $y^{-1}$ (O'Doherty et al., 2014). In estimation of $RE_{south}$, small value of $RE_{south}$ would be deduced from difference in the rates of increase of atmospheric concentrations of $CH_2F_2$ between hemispheres. Dissolution of $CH_2F_2$ in the ocean in the Southern Hemisphere may thus affect estimation of $RE_{south}$ and then affect estimation of $CH_2F_2$ emissions in the Southern Hemisphere and their seasonal variability.

I will revise the text as follows.

L26-29, P10:

Hence, dissolution of $CH_2F_2$ in the ocean, even if dissolution is reversible, may influence estimates of $CH_2F_2$ emissions derived from long-term observational data on atmospheric concentrations of $CH_2F_2$; in particular, consideration of dissolution of $CH_2F_2$ in the ocean may affect estimates of $CH_2F_2$ emissions in the Southern Hemisphere and their seasonal variability because of slow rates of inter-hemispheric transport and small portion of the $CH_2F_2$ emissions in the Southern Hemisphere to the total emissions.

**Specific comments:**

**P2, L28: 'first' is written twice.**

Thank you for the comment.
The text is revised as follows.

First, the values of $K_H$ for $CH_2F_2$ were determined over the temperature range from 276 to 313 K by means of an inert-gas stripping (IGS) method.

**P5, L8: add errors for numbers. Add typical values, their units, and uncertainties of variables for the key equations throughout the manuscript.**

Thank you for the comment.

If redistribution of $CH_2F_2$ in the headspace to the test solution had occurred, the $K_H$ values determined in this study would be overestimated. Errors due to this redistribution are always negative values. The ratio of the errors to the $K_{eq}$ values (%) is $100 \times \frac{\left(\frac{V_{head}}{RTV}\right)}{\left(\frac{1}{k_1 RTV} F\right)}$, that is, $\frac{100 k_1 V_{head}}{F}$. Under the experimental conditions here, this ratio is calculated to be $-2.0$ to $-2.3$ % at 3.0 °C and $-4.6$ to $-5.1$ % at 39.5 °C. Values of this ratio increase as values of $K_{eq}$ decrease. This ratio is maximum ($-6.5$ %) for a-seawater at 51.534‰ and 39.5 °C.

Typical values, their units, and uncertainties of variables for the key equations are added as follows.

L6-8, P4:

[revised manuscript text omitted]

**P7, L6: What statistical test for outliers was applied? How many points were removed at each temperature.**

Thank you for the comment.

Statistical test for outliers is as follows.

The data with errors being >10% of the data was first excluded. Next, some data were excluded for calculation of the average so that the remaining data were inside the $2\sigma$ range. This procedure was iterated until all the data were inside the $2\sigma$ range.

The data points thus excluded was only for $V$ values of 0.350 dm$^3$. The number of them were eight or fewer at each temperature. The maximum number of the data excluded was corrected to be eight although it was described to be six in the original manuscript. Number of the data thus excluded were indicated in Tables 1m (shown in R1) and 3m.

**Table 3m.** The average of values of $F/(k_1RTV)$ obtained for $V$ value of 0.350 dm$^3$ and the $K_{eq}^S(T)$ value derived from Eq. (23) at each salinity and temperature. $N$ represents number of experimental runs for the average.

| | $K_{eq}^S$ (M atm$^{-1}$) | | | | | |
|---|---|---|---|---|---|---|
| $T$ (K) | salinity, 4.452 ‰ | | | salinity, 8.921 ‰ | | |
| | average [a, b] | $N$ [c] | Eq. (23) | average [a, b] | $N$ [c] | Eq. (23) |
| 276.15 | 0.108 ± 0.006 (0.008) | 8 (0) | 0.108 | 0.103 ± 0.006 (0.008) | 21 (0) | 0.104 |
| 278.35 | 0.099 ± 0.004 (0.006) | 12 (0) | 0.099 | 0.095 ± 0.006 (0.008) | 26 (1) | 0.095 |
| 283.65 | 0.086 ± 0.003 (0.005) | 9 (0) | 0.085 | 0.083 ± 0.007 (0.009) | 24 (0) | 0.082 |
| 288.65 | 0.075 ± 0.004 (0.006) | 12 (0) | 0.074 | 0.072 ± 0.005 (0.006) | 33 (0) | 0.071 |
| 293.45 | 0.065 ± 0.002 (0.003) | 10 (0) | 0.065 | 0.063 ± 0.003 (0.004) | 27 (5) | 0.062 |
| 298.15 | 0.058 ± 0.002 (0.003) | 13 (0) | 0.058 | 0.056 ± 0.004 (0.005) | 26 (2) | 0.056 |
| 303.05 | 0.052 ± 0.001 (0.002) | 8 (0) | 0.052 | 0.049 ± 0.004 (0.005) | 14 (6) | 0.050 |
| 307.95 | 0.047 ± 0.002 (0.003) | 13 (1) | 0.047 | 0.046 ± 0.004 (0.005) | 23 (1) | 0.045 |
| 312.65 | 0.042 ± 0.001 (0.002) | 7 (0) | 0.042 | 0.040 ± 0.003 (0.004) | 12 (8) | 0.041 |

| | $K_{eq}^S$ (M atm$^{-1}$) | | | | | |
|---|---|---|---|---|---|---|
| $T$ (K) | salinity, 21.520 ‰ | | | salinity, 36.074 ‰ | | |
| | average [a, b] | $N$ [c] | Eq. (23) | average [a, b] | $N$ [c] | Eq. (23) |
| 276.15 | 0.095 ± 0.006 (0.008) | 20 (0) | 0.095 | 0.088 ± 0.005 (0.007) | 21 (0) | 0.088 |
| 278.35 | 0.087 ± 0.005 (0.007) | 22 (0) | 0.087 | 0.079 ± 0.006 (0.008) | 20 (3) | 0.081 |
| 283.65 | 0.075 ± 0.004 (0.006) | 15 (1) | 0.076 | 0.069 ± 0.002 (0.003) | 18 (2) | 0.071 |
| 288.65 | 0.066 ± 0.004 (0.005) | 20 (0) | 0.066 | 0.062 ± 0.004 (0.005) | 19 (4) | 0.062 |
| 293.45 | 0.058 ± 0.003 (0.004) | 14 (0) | 0.058 | 0.054 ± 0.002 (0.003) | 19 (4) | 0.055 |
| 298.15 | 0.052 ± 0.003 (0.004) | 20 (0) | 0.052 | 0.049 ± 0.002 (0.003) | 24 (4) | 0.049 |
| 303.05 | 0.046 ± 0.003 (0.004) | 16 (0) | 0.046 | 0.044 ± 0.002 (0.003) | 16 (0) | 0.044 |
| 307.95 | 0.042 ± 0.003 (0.004) | 16 (0) | 0.042 | 0.040 ± 0.002 (0.003) | 15 (2) | 0.040 |
| 312.65 | 0.038 ± 0.002 (0.003) | 16 (0) | 0.038 | 0.036 ± 0.002 (0.003) | 16 (0) | 0.037 |

| | $K_{eq}^S$ (M atm$^{-1}$) | | |
|---|---|---|---|
| $T$ (K) | salinity, 51.534 ‰ | | |
| | average [a, b] | $N$ [c] | Eq. (23) |
| 276.15 | 0.081 ± 0.003 (0.005) | 10 (0) | 0.082 |
| 278.35 | 0.077 ± 0.003 (0.005) | 15 (0) | 0.076 |
| 283.65 | 0.067 ± 0.001 (0.003) | 9 (1) | 0.066 |
| 288.65 | 0.059 ± 0.002 (0.003) | 14 (1) | 0.058 |
| 293.45 | 0.052 ± 0.001 (0.002) | 7 (3) | 0.052 |
| 298.15 | 0.047 ± 0.002 (0.003) | 15 (0) | 0.047 |
| 303.05 | 0.042 ± 0.001 (0.002) | 8 (0) | 0.042 |
| 307.95 | 0.038 ± 0.002 (0.003) | 12 (0) | 0.039 |
| 312.65 | 0.036 ± 0.001 (0.002) | 7 (1) | 0.036 |

a. Errors are 2σ for the average only.; b. Number in parenthesis represents an error reflecting both 2σ for the average and potential systematic bias.; c. Number in parenthesis represents number of experimental runs excluded for the average.

**Eq (17). For sake of discussion, can a $k_S$ value be given here? And what is the effect of including $k_S$ vs $k_{S1}$, $k_{S2}$ in the model - does it make a difference?**

Thank you for the comment.

In Table S7, values of $k_s$ were given, and the $K_{eq}$ values calculated by Eq. (17) were compared to those by Eq. (22) at salinity of 30, 35 and 40 ‰ and each temperature. The text on lines 6-8 and lines 20-21 in page 9 will be revised as follows.

Figure 5 plots $\ln(K_H(T)/K_{eq}^S(T))$ against $S$ at each temperature. Table S7 lists values of $k_s$ determined by fitting the data at each temperature by use of Eq. (17). If the $K_{eq}^S(T)$ values obeyed Eq. (17), the data at each temperature in Fig. 5 would fall on a straight line passing through the origin, but they did not. Figure 5 reveals that the salinity dependence of $CH_2F_2$ solubility in a-seawater cannot be represented by Eq. (17).

$\ln(K_H(T)/K_{eq}^S(T))$ depends on $S^{0.5}$ and follows Eq. (22) rather than the Setchenow equation (Eq. (17)). Table S14m lists ratios of $K_{eq}^S$ calculated by Eq. (17) to those by Eq. (22). Difference between values of $K_{eq}^S$ calculated by Eqs. (17) and (22) at 35‰ of salinity was within 3% of the $K_{eq}^S$ value. Decreases in values of $K_{eq}^S$ are calculated to be 7–8% and 4%, respectively, by Eqs. (17) and (23) as salinity of a-seawater increases from 30‰ to 40‰ at each temperature.

The reason for this salting-out effect of $CH_2F_2$ solubility in a-seawater is not clear.

**Table S7. Values of $k_s$ (Eq. (17)) and comparison of values of $K_{eq}^S$ calculated at each temperature by Eq. (17) with those by Eq. (22).**

| Temperature (°C) | $k_s$ (‰$^{-1}$) | $[K_{eq}^S$ from Eq. (17)]/ $[K_{eq}^S$ from Eq. (22)] | | | $[K_{eq}^S$ at 30‰]/ $[K_{eq}^S$ at 40‰] | |
|---|---|---|---|---|---|---|
| | | at 30‰ | at 35‰ | at 40‰ | Eq. (17) | Eq. (23) |
| 3.0 | 0.00811 | 1.027 | 1.008 | 0.988 | 1.084 | 1.043 |
| 5.8 | 0.00785 | 1.033 | 1.014 | 0.995 | 1.082 | 1.042 |
| 10.5 | 0.00768 | 1.033 | 1.016 | 0.997 | 1.080 | 1.042 |
| 15.5 | 0.00718 | 1.044 | 1.028 | 1.012 | 1.074 | 1.041 |
| 20.3 | 0.00728 | 1.037 | 1.020 | 1.003 | 1.076 | 1.040 |
| 25.0 | 0.00704 | 1.040 | 1.024 | 1.008 | 1.073 | 1.039 |
| 29.9 | 0.00731 | 1.027 | 1.010 | 0.992 | 1.076 | 1.039 |
| 34.8 | 0.00713 | 1.029 | 1.012 | 0.995 | 1.074 | 1.038 |
| 39.5 | 0.00709 | 1.026 | 1.010 | 0.992 | 1.073 | 1.038 |

---

## Author Response (AR2)

**Response to the comments by Co-Editor, Referee 1 and Referee 2.**

According to Co-Editor's comments, the spelling is changed from "Setchenow" to "Sechenov" through the manuscript.

The response to Referee 1 and Referee 2 follows sequence: (1) comments from Co-Editor and Referees, (2) author's response, (3) author's changes in manuscript.

The author's changes are marked in blue. In addition, I provide a marked-up manuscript version showing the changes made (using track changes in Word).

I will reply to each comment as follows.

**1. To the comments by Referee 1:**

Thank you very much for the constructive comments.

I will reply to each comment as follows.

**R1-1** ---------------------------------------------------------------------------------------------------------------------------

(1) comments from Referee 1

A glance at Figure 2 gives the reassuring impression that random errors are quite small for both the IGS and PRV-HS methods. However, there is a small but significant difference in the results at the one temperature where both techniques are used. This shows a systematic error in one or both of the methods.

(2) author's response

Thank you for the comment. Evaluation of systematic errors or potential systematic bias is also commented by Referee 2. Potential systematic bias of values of $K_{eq}$ determined are estimated to be within ±2% in the IGS method and within ±4% in the PRV-HS method, as described in the revised manuscript.

In the original manuscript, error bars in Fig. 2 represent statics errors (*error_S*) only. I revise Fig. 2 by plotting the data with error bars (*error_T*) representing both *error_S* and potential systematic bias (*error_B*). Values of *error_T* are calculated by (*error_S* + *error_B*) rather than $\sqrt{(error\_S)^2 + (error\_B)^2}$ because *error_B* is potential systematic bias. Tables 1, S1 and 3 list *error_T* as well as *error_S* in the revised manuscript. Fig. 4 is also revised by plotting the data with error bars (*error_T*). Error bars of values of $\frac{F}{k_1 RTV}$ in Figs. 1, 3, S5, S6, S7, and S8 are explained in the captions.

As seen in Fig. 2 in the revised manuscript, potential systematic bias in both the PRV-HS method and the IGS method could be a reason why there is the small offset between PRV-HS and IGS method at 312 K

(3) author's changes in manuscript

lines 13-17, page 5

[revised manuscript text omitted]

a. Errors are 2σ for the regression only.; b. Errors are those at 95% confidence level for the regression only.; c. Number in parenthesis represents both errors at 95% confidence level for the regression and potential systematic bias (±4%).

[revised manuscript text omitted]

**R1-2** ----------------------------------------------------------------------------------------------------------------------

(1) comments from Referee 1

In the fitting equation, equation 13, the number of significant figures reported is much higher than justified for the relatively small number of data points. In nonlinear fitting of this type, most programs report the variance associated with each of the fitting coefficients. If the square-root-of-variance is not small compared to the fitting coefficient, that means that the inclusion of that coefficient is probably not justified.

(2) author's response

Thank you for the comments. According to Referee 1's comment, I revise the significant figure of each fitting coefficient in Eq. (13). I set the least digit of the significant figure to the second decimal place so that the values calculated by Eq. (13) are consistent with the significant figure of $K_H$.

Thank you for the suggestion that the square-root-of-variance of the fitting coefficient should be checked for justifying whether the coefficient should be included in the van't Hoff equation.

The square-root-of-variance, that is, standard deviation for each fitting coefficient in Eq. (12) justifies the three-term van't Hoff equation. The standard deviation for each fitting coefficient is described in the revised manuscript. Because the ratio of $2 \times \delta a_3/a_3$ is 0.293, the three-term van't Hoff equation is thus justified.

In addition, even if the data only in the IGS method is fitted separately, a three-term fit to the data in the IGS method would be justified as Eq. (A1), although errors of the fitting coefficients are larger than those in Eq. (13).

$$\ln(K_H(T)) = (-41.7 \pm 7.2) + (66.8 \pm 10.5) \times \left(\frac{100}{T}\right) + (15.1 \pm 3.7) \times \ln\left(\frac{T}{100}\right) \tag{A1}$$

where errors of the fitting coefficients represent standard deviation only for non-linear fitting.

(3) author's changes in manuscript

lines 14-16, page 8

$$\ln(K_H(T)) = -49.71 + 77.70 \times \left(\frac{100}{T}\right) + 19.14 \times \ln\left(\frac{T}{100}\right) \tag{13.}$$

The square-root-of-variance, that is, standard deviation for each fitting coefficient in Eq. (12) is as follows:

$\delta a_1 = 5.5$; $\delta a_2 = 8.3$; $\delta a_3 = 2.8$.

**R1-3** ----------------------------------------------------------------------------------------------------------------------

(1) comments from Referee 1

The treatment of the salting-out effect is overworked. In Referee 1's opinion, lines 9-26, page 9 should be eliminated and the author should simply state that $\ln(K_H/K_{eq})$ varies close to the 0.5 power of salinity, in contrast to the Sechenov.

(2) author's response

   Thank you for the constructive comments. I agree the comment that the treatment of the salting-out effect is overworked. I followed the referee's opinion and found that all the data in Fig. 5 could be fitted using only one parameter (Eq. (22)) as described in the revised manuscript.

5    In the revised manuscript, error bars in Fig. 5 reflect *error_T* (R1-1) and Figs. 4 (shown in R1-2) and 6 are redrawn according to Eq. (23).

(3) author's changes in manuscript

lines 10-12, page 1, Abstract

10  The salinity dependence of $K_{eq}^S$ (the salting-out effect), $\ln(K_H/K_{eq}^S)$, did not obey the Sechenov equation but was proportional to $S^{0.5}$. Overall, the $K_{eq}^S(T)$ value was expressed by $\ln(K_{eq}^S(T)) = -49.71 + (77.70 - 0.134 \times S^{0.5}) \times (100/T) + 19.14 \times \ln(T/100)$.

lines 2-17, page 10

15     This result suggests that $\ln(K_H(T)/K_{eq}^S(T))$ varied according to Eq. (18):

   $\ln(K_H(T)/K_{eq}^S(T)) = k_{s1}\ S^{0.5}$                                     (18)

   Values of $k_{s1}$ may be represented by the following function of $T$:

   $k_{s1} = b_1 + b_2 \times (100/T)$                                            (19)

   Parameterizations of $b_1$ and $b_2$ obtained by fitting all the $\ln(K_H(T)/K_{eq}^S(T))$ and $S$ data at each temperature simultaneously by

20  means of the nonlinear least-squares method gives Eq. (20).

   $\ln(K_H(T)/K_{eq}^S(T)) = (0.0127 + 0.0099 \times (100/T)) \times S^{0.5}$       (20)

   The standard deviation for each fitting coefficient in Eq. (19) is as follows:

      $\delta b_1 = 0.0106;\ \delta b_2 = 0.0031$.

   Since $2 \times \delta b_1 > b_1$, the parameterization by Eq. (19) may be overworked. Accordingly, all the $\ln(K_H(T)/K_{eq}^S(T))$ and $S$ data at

25  each temperature are fitted simultaneously using Eq. (21) instead of Eq. (19). The nonlinear least-squares method gives Eq. (22).

   $k_{s1} = b_2 \times (100/T)$                                                 (21)

   $\ln(K_H(T)/K_{eq}^S(T)) = 0.134 \times (100/T) \times S^{0.5}$               (22)

   The standard deviation for the fitting coefficient in Eq. (21) is as follows: $\delta b_2 = 0.001$. As seen in Fig. 5, Eqs. (21) and (22)

30  reproduced the data well.

lines 25-26, page 10

In Eq. (22), $K_H(T)$ is represented by Eq. (13), as described in Sect. 3.1. Therefore $K_{eq}^{S}(T)$ is represented by Eq. (23):

$$\ln\left(K_{eq}^{S}\right) = -49.71 + (77.70 - 0.134 \times S^{0.5}) \times \left(\frac{100}{T}\right) + 19.14 \times \ln\left(\frac{T}{100}\right) \tag{23}$$

Fig 6

[Figure]

**Figure 6. Plots of monthly averaged equilibrium fractionation of $CH_2F_2$ between atmosphere and ocean, $R_m$ (Gg patm$^{-1}$) in the global and the semi-hemispheric atmosphere. Right vertical axis represents monthly averaged residence ratio of $CH_2F_2$ dissolved in the ocean mixed layer to the atmospheric burden for each lower tropospheric semi-hemisphere of the AGAGE 12-box model.**

**R1-4** ----------------------------------------------------------------------------------------------------------------------------------

(1) comments from Referee 1

On page 3, lines11-12 (in the revised manuscript), the water quality should be indicated as (resistivity > 18 megohm-cm).

(2) author's response

Thank you for the comment. I correct the text according to the comment.

(3) author's changes in manuscript

Water was purified with a Milli-Q Gradient A10 system (resistivity > 18 megohm-cm).

**R1-5** ------------------------------------------------------------------------------------------------------------------

5  (1) comment from Referee 1

On page 7, line 1 (in the revised manuscript), "non-linear" is misspelled.

(2) author's response

Thank you for the comment. I correct the text according to the comment.

(3) author's changes in manuscript

lines 1-2, page 7

Furthermore, values of $K_{eq}(T)$ and errors of them were determined by nonlinear fitting of the data of $L_i$ and $V_i/V$ by means of Eq. (11), which was obtained from Eq. (10):

**2. To the comments by Referee 2:**

Thank you very much for the constructive comments.

I will reply to each comment as follows.

**R2-1** ------------------------------------------------------------------------------------------------------------------

(1) comments from Referee 2

The manuscript would benefit from an explicit discussion of experimental error. What are the parameters that limit the accuracy of the inert-gas stripping (IGS) method? Of the stripping column apparatus? And of the phase

25  ratio variation headspace method (PRV-HS)? Do error bars reflect statics only, or also potential sources of systematic bias?

(2) author's response

Thank you for the comment. I reply to the comments in the sequence: (i) on the parameters that limit the

30  accuracy of the IGS method; (ii) on the parameters that limit the accuracy of the PRV-HS method; and (ii) on error bars.

(i) the parameters that limit the accuracy of the IGS methods

The parameters that limit the accuracy of the IGS methods are temperature of the test solution ($T$) and flow rate of purge gas ($F$).

The accuracy of $T$ ($\delta T$) are within 0.2 K and may give potential systematic bias of ±0.5 to ±0.6 % ($\delta K_{eq}/K_{eq}$), where $\delta K_{eq}$ indicates an error of $K_{eq}$.

5 For $F$, the accuracy of $F_{meas}$ is estimated to be within 1% from the accuracy of the high-precision film flow meter SF-1U with VP-2U used for calibrating the soap flow meter. Errors in the term of $\frac{P_{meas}-h_{meas}}{P_{hs}-h} \times \frac{T}{T_{meas}}$ in Eq (3) are estimated at ca. ±1 %. Hence, the accuracy of $F$ ($\delta F$) are estimated to be within 1.4 % and may give potential systematic bias of ±1.4 % of $\delta K_{eq}/K_{eq}$.

Values of $\delta K_{eq}/K_{eq}$ due to both $\delta T$ and $\delta F$ may thus have potential systematic bias of ca. ±2%.

(ii) the parameters that limit the accuracy of the PRV-HS methods

The parameters that limit the accuracy of the PRV-HS methods are temperature of the test solution ($T$) and volume of the vials used ($V$).

Although the apparatus used (Agilent, HP7694) was expected to keep $T$ constant, the accuracy of $T$ may not
15 be certified. I have applied the same apparatus to determination of the $K_H$ values for some HCFCs such as HCFC-123 using the PRV-HS methods [Kutsuna, S. *Int. J. Chem. Kinet.*, 45, 440-451, 2013]. On the basis of the $K_H$ values thus determined and comparison between them and the reported values for HCFC-123, errors of $T$ are estimated to be within ca. 2 K. These errors of $T$ may give potential systematic bias of ca. ±4 % ($\delta K_{eq}/K_{eq}$) at 313 K and ca. ±3 % ($\delta K_{eq}/K_{eq}$) at 353 K.

20 Errors for $V$ ($\delta V$) are estimated to be less than 1 %, and these errors may give potential systematic bias of less than 1 % of $\delta K_{eq}/K_{eq}$.

Accordingly, for the PRV-HS methods, values of $\delta K_{eq}/K_{eq}$ due to both $\delta T$ and $\delta V$ may have potential systematic bias of ca. ±4%.

25 (iii) Error bars in Figure 2

Error bars in Figure 2 reflect statics only (*error_S*) in the original manuscript. Error bars in Figure 2m represent errors (*error_T*) reflecting both *error_S* and potential systematic bias (*error_B*). Values of *error_T* are also indicated in Tables 1m, S1m and 3m. Values of *error_T* are calculated by (*error_S* + *error_B*) rather than $\sqrt{(error\_S)^2 + (error\_B)^2}$ because *error_B* is potential systematic bias.

(3) author's changes in manuscript

The change in manuscript is the same as described in R1-1.

**R2-2** --------------------------------------------------------------------------------------------------------------------------

(1) comments from Referee 2

   There does appear to be a small -yet significant- offset between PRV-HS and IGS method in Figure 2. Why IGS is believed to be more accurate?

(2) author's response

   Thank you for the comment. There appears to be a small - yet significant - offset between PRV-HS and IGS method at 312 K. This point is also commented by Referee 1. For the PRV-HS methods, values of $\delta K_{eq}/K_{eq}$ may have potential systematic bias of ca. ±4%, which results mostly from the accuracy of temperature of the test

10 solution, as aforementioned (R2-1). For the IGS method, values of $\delta K_{eq}/K_{eq}$ may have potential systematic bias of ca. ±2%. The IGS method is thus believed to be more accurate. Potential systematic bias in both the PRV-HS method and the IGS method could be a reason why there is the small offset between PRV-HS and IGS method at 312 K.

15 (3) author's changes in manuscript

   The change in manuscript related to this comment is included in the changes described in R1-1.

**R2-3** --------------------------------------------------------------------------------------------------------------------------

(1) comments from Referee 2

20 Does the fit according to (Eq (13)) take into account the relative weight of error bars?

(2) author's response

   Thank you for the comment. The fit according to Eq. (13) does not take into account the relative weight of error bars. This is clearly described in the revised manuscript.

(3) author's changes in manuscript

lines 10-11, page 8:

   All the $K_H$ values were regressed with respect to the van't Hoff equation (Eq. (12)) with no weighting (Clarke and Glew, 1965; Weiss, 1970):

**R2-4** --------------------------------------------------------------------------------------------------------------------------

(1) comments from Referee 2

   What is the reason for the large variation in the size of error bars in Fig. 5?

(2) author's response

  Thank you for the comment. As Referee 2 comments, there are the large variation in the size of error bars in Fig. 5. Ratio among error bars of the data at the same temperature is up to maximum value of 4.5: error bars are 0.084 for 8.921‰ and 0.019 for 51.534‰ at 10.5 °C. Error bars for the data at 8.921‰ tend to be large and error

5 bars for the data at 51.534‰ tend to be small: this reflects statics errors of the data at 8.921‰ and 51.534‰.

  Errors of the data in Fig. 5 represents statics only (error_S, as shown in R2-1). As replied in R2-1, errors from both statics (error_S) and potential systematic bias of ±2% (error_B) will be used as errors (error_T) for the data in Fig. 5: (error_T) = (error_S) + (error_B). In the revised manuscript, error bars of the data in Fig. 5 represent error_T. As seen in Fig. 5 in the revised manuscript, the ratios among error bars of the data at the

10 same temperature are smaller than the corresponding ratios in Fig. 5 in the original manuscript. For example, the ratio of error bars between at 8.921‰ and 51.534‰ at 10.5 °C is 2.7 while it is 4.5 in the original manuscript as aforementioned.

  In the revised manuscript, error bars will be represented by error_T in Fig. 5.

15 (3) author's changes in manuscript
Figure 5:

[Figure]

**Figure 5.** Plots of $\ln(K_H(T)/K_{eq}{}^S(T))$ vs. salinity in a-seawater at each temperature. Bold curves represent the fitting obtained by Eq. (22). Error bars represent errors reflecting both $2\sigma$ for the average and potential systematic bias (±2%) of $K_{eq}{}^S$.

20 **R2-5** ---------------------------------------------------------------------------------------------------------------------

**(1) comments from Referee 2**

The $S^{0.5}$ components of the fit (deviation from Sechenov) is strongest at warm temperatures, and smallest at low temperatures. This is an interesting observation, that warrants discussion. What are possible causes? What is its relevance?

**(2) author's response**

Thank you for the comment. The reason why $\ln(K_H/K_{eq}{}^S)$ is proportional to $S^{0.5}$ rather than $S$ is still unclear. I will describe a potential reason for this proportionality simply in the text, and make discussion in *Supporting Information*.

**(3) author's changes in manuscript**

lines 22-24, page 10:

The reason for this salting-out effect of $CH_2F_2$ solubility in a-seawater is not clear. Specific properties of $CH_2F_2$ —small molecular volume, which results in small work of cavity creation (Graziano, 2004; 2008), and large solute-solvent attractive

15  potential energy in water and a-seawater— may cause deviation from Sechenov relationship (Sect. S5, *Supporting Information*).

Sect. S5, 2nd block, page 8 - page 9 in Supporting Information:

I calculate Ben-Naim standard Gibbs energy $\Delta G^{\bullet}$, enthalpy $\Delta H^{\bullet}$, and entropy $\Delta S^{\bullet}$ changes for dissolution of $CH_2F_2$ in

20  water because these values correspond to the values for the transfer from a fixed position in the gas phase to a fixed position in water. Values of $\Delta G^{\bullet}$, $\Delta H^{\bullet}$, and $\Delta S^{\bullet}$ are calculated on the basis of the Ostwald solubility coefficient, $L(T)$, as follows.

$$\ln(L(T)) = \ln\left(RTK_{eq}{}^S(T)\right) \tag{B1}$$

$$\Delta G^{\bullet} = R'T\ln(L(T)) \tag{B2}$$

$$\Delta H^{\bullet} = -\frac{\partial}{\partial(^1/_T)}\left(\frac{\Delta G^{\bullet}}{T}\right) \tag{B3}$$

25  $$\Delta S^{\bullet} = \frac{\Delta H^{\bullet} - \Delta G^{\bullet}}{T} \tag{B4}$$

where both $R$ and $R'$ represent gas constant but their units are different: $R = 0.0821$ in atm dm$^3$ K$^{-1}$ mol$^{-1}$; $R' = 8.314$ in J K$^{-1}$ mol$^{-1}$.

Combining Eqs. (B1), (B2), (B3), and (B4) with Eqs. (14) and (15), $\Delta G^{\bullet}$ (kJ mol$^{-1}$), $\Delta H^{\bullet}$ (kJ mol$^{-1}$), and $\Delta S^{\bullet}$ (J mol$^{-1}$ K$^{-1}$) are represented by $\Delta G_{sol}$ and $\Delta H_{sol}$ as follows:

30  $$\Delta G^{\bullet} = \Delta G_{sol} + R'T\ln(RT) \tag{B5}$$

$$\Delta H^{\bullet} = \Delta H_{sol} + R'T \tag{B6}$$

$$\Delta S^{\bullet} = \frac{\Delta H_{sol} - \Delta G_{sol}}{T} + R' - R'\ln(RT) \tag{B7}$$

Values of $\Delta G^{\cdot}$, $\Delta H^{\cdot}$, and $\Delta S^{\cdot}$ calculated at 298 K are listed in Table S2. Table S2 also lists values of $\Delta G^{\cdot}$, $\Delta H^{\cdot}$, and $\Delta S^{\cdot}$ reported for CH$_3$F and C$_2$H$_6$ (Graziano, 2004) and CH$_4$ (Graziano, 2008) at 298 K. The chemicals, which having a methyl group, in Table S2 are classified into two groups (CH$_2$F$_2$ and CH$_3$F; CH$_4$ and C$_2$H$_6$) according to $\Delta G^{\cdot}$.

Table S2 lists values of $\Delta G_c$, $E_a$ and $\Delta H^h$ deduced using a scaled particle theory (Granziano, 2004; 2008). $\Delta G_c$ is the

5  work of cavity creation to insert a solute in a solvent. $E_a$ is a solute-solvent attractive potential energy and accounts for the solute-solvent interactions consisting of dispersion, dipole-induced dipole, and dipole-dipole contributions. $\Delta H^h$ is enthalpy of solvent molecules reorganization caused by solute insertion. The solvent reorganization mainly involves a rearrangement of H-bonds.

$\Delta G_c$ is entropic in nature in all liquids, being a measure of the excluded volume effect due to a reduction in the spatial

10  configurations accessible to liquid molecules upon cavity creation. Hence, C$_2$H$_6$ has larger value of $\Delta G_c$ than CH$_3$F and CH$_4$. $\Delta G_c$, $E_a$, and $\Delta H^h$ are related to $\Delta G^{\cdot}$ and $\Delta H^{\cdot}$ as follows (Graziano, 2008):

$$\Delta G^{\cdot} = \Delta G_c + E_a \tag{B8}$$

$$\Delta H^{\cdot} = E_a + \Delta H^h \tag{B9}$$

Table S3 thus suggests that smaller value of $\Delta G^{\cdot}$ of CH$_3$F than CH$_4$ is due to large solute-solvent attractive potential energy

15  ($-E_a$) of CH$_3$F.

**Table S3. Ben-Naim standard hydration Gibbs energy $\Delta G^{\cdot}$, enthalpy $\Delta H^{\cdot}$, and entropy $\Delta S^{\cdot}$ changes for dissolution of CH$_2$F$_2$ at 298 K determined here and the corresponding values and values of $\Delta G_c$, $E_a$ and $\Delta H^h$ reported for CH$_3$F and C$_2$H$_6$ (Granziano, 2004) and CH$_4$ (Graziano, 2008).**

|  | $\Delta G^{\cdot}$ (kJ mol$^{-1}$) | $\Delta H^{\cdot}$ (kJ mol$^{-1}$) | $\Delta S^{\cdot}$ (J K$^{-1}$ mol$^{-1}$) | $\Delta G_c$ (kJ mol$^{-1}$) | $E_a$ (kJ mol$^{-1}$) | $\Delta H^h$ (kJ mol$^{-1}$) |
|---|---|---|---|---|---|---|
| CH$_2$F$_2$ | −1.1 | −14.7 | −45.4 |  |  |  |
| CH$_3$F | −0.9 | −15.8 | −50.0 | 23.3 | −24.3 | 8.5 |
| CH$_4$ | 8.4 | −10.9 | −64.7 | 22.9 | −14.5 | 3.7 |
| C$_2$H$_6$ | 7.7 | −17.5 | −84.5 | 28.4 | −20.7 | 3.2 |

Graziano (2008) definitively explained the salting-out of CH$_4$ by sodium chloride at molecular level on the basis of a scaled particle theory. He explained that $\Delta G_c$ increase was linearly related to the increase in the volume packing density of the solutions ($\xi_3$) with adding NaCl. Such an increase of $\Delta G_c$ is probably the case for salting-out of CH$_2$F$_2$ by a-seawater observed in this study. He also explained that $E_a$ was linearly related to the increase in $\xi_3$ assuming that a fraction of the

25  dipole-induced dipole attractions could be taken into account by the parameterization of the dispersion contribution.

I think the possibility that $E_a$ may be nonlinearly related to the increase in $\xi_3$ because of dipole-dipole interaction between CH$_2$F$_2$ and solvents. Temperature dependence in Eq. (22) suggests that salting-out effect of CH$_2$F$_2$ by a-seawater is enthalpic. Eqs. (22) and (B9) thus suggests that the salting-out of CH$_2$F$_2$ is mostly related to change in $E_a$. CH$_2$F$_2$ has relatively small value of $\Delta G_c$ because of its small molecular volume compared to other chemicals such as C$_2$H$_6$. Accordingly,

$\Delta G^{\cdot}$, that is, solubility of $CH_2F_2$ would depend on $E_a$ rather than $\Delta G_c$. Therefore, I think that specific properties of $CH_2F_2$ – small molecular volume, which results in small work of cavity creation (Graziano, 2004; 2008), and large solute-solvent attractive potential energy in water and a-seawater– may cause deviation from Sechenov relationship.

5    The following two references will be cited both in the manuscript and in *Supporting Information*.

Graziano, G.: Case study of enthalpy–entropy noncompensation. Journal of Chemical Physics, 120, 4467-4471, doi: 10.1063/1.1644094, 2004.

Graziano, G.: Salting out of methane by sodium chloride: A scaled particle theory study. Journal of Chemical Physics, 129, 084506, doi: 10.1063/1.2972979, 2008.

**R2-6** ----------------------------------------------------------------------------------------------------------------------

(1) comments from Referee 2

The discussion in Sect. 3.3 assumes solubility equilibrium with the atmosphere over the full depth of the ocean mixed layer. How deep is this mixed ocean layer in the model? Does this mean the model estimates an upper
15    limit?

(2) author's response

Thank you for the comments. The depth of the ocean mix layer in the model is 10 to 600 m. The depth distribution of $CH_2F_2$ dissolved in the ocean mixed layer in each semi-hemisphere is listed in Tables S4 (30° S–
20    90° S), S5 (30° S–0° S), S6 (0° N–30° S) and S7 (30° N–90° N) in the revised manuscript. As seen in these tables, the $CH_2F_2$ dissolved in the ocean mixed layer resides mostly in less than 300 m depth. For example, for the southern semi-hemisphere (30° S–90° S) (Table S4), in August, when the amount of $CH_2F_2$ dissolved in the ocean mixed layer is maximum, 66% of the $CH_2F_2$ would be dissolved in the mixed layer with its depth between 100 m and 200 m, and 91 % of the $CH_2F_2$ dissolved in the ocean mixed layer is expected to reside in less than
25    300 m depth.

As Referee 2 pointed out, model estimates mean an upper limit of the amount of $CH_2F_2$ dissolved in the ocean mixed layer. This point will be clearly described in the revised manuscript.

(3) author's changes in manuscript
30    line 29, page 11 - line 11, page 12:

As seen in Figure 6, in the southern semi-hemispheric lower troposphere (30° S–90° S), at least 5 % of the atmospheric burden of $CH_2F_2$ would reside in the ocean mixed layer in the winter, and the annual variance of the $CH_2F_2$ residence ratio would be 4%. These ratios are, in fact, upper limits because $CH_2F_2$ in the ocean mixed layer may be undersaturated. It takes days to a few weeks after a change in temperature or salinity for oceanic surface mixed layers to come to equilibrium with

the present atmosphere, and equilibration time increases with depth of the surface mixed layer (Fine, 2011). In the estimation using the gridded data here, >90 % of $CH_2F_2$ in the ocean mixed layer would reside in less than 300 m depth (Tables S4, S5, S6 and S7).

Haine and Richards (1995) demonstrated that seasonal variation in ocean mixed layer depth was the key process which

5    affected undersaturation and supersaturation of chlorofluorocarbon 11 (CFC-11), CFC-12 and CFC-113 by use of a one-dimensional slab mixed model. As described above, >90 % of $CH_2F_2$ in the ocean mixed layer is expected to reside in less than 300 m depth. According to the model calculation results by Haine and Richards (1995), saturation of $CH_2F_2$ would be >0.9 for the ocean mixed layer with less than 300 m depth. The saturation of $CH_2F_2$ in the ocean mixed layer is thus estimated to be at least 0.8. In the southern semi-hemispheric lower troposphere (30° S–90° S), therefore, at least 4 % of the

10    atmospheric burden of $CH_2F_2$ would reside in the ocean mixed layer in the winter, and the annual variance of the $CH_2F_2$ residence ratio would be 3%.

The following two references will be cited in the manuscript.

Fine, R. A.: Observations of CFCs and $SF_6$ as ocean tracers. Annual Review of Marine Science, 3, 173-195,

15    doi:10.1146/annurev.marine.010908.163933, 2011.

Haine, T. W. N. and Richards, K. J.: The influence of the seasonal mixed layer on oceanic uptake of CFCs. Journal of Geophysical Research, 100, 10727-10744, doi:10.1029/95JC00629, 1995.

Supporting Information, Tables S4, S5, S6 and S7

20    **Table S4. Monthly amount of $CH_2F_2$ dissolved in the ocean mixed layer at solubility equilibrium with the atmospheric $CH_2F_2$ (partial pressure, 1 patm) and the depth distribution of the $CH_2F_2$ dissolved in the southern semi-hemisphere (90°S - 30°S).**

| | Amount (Gg patm$^{-1}$) | Distribution of the amount of $CH_2F_2$ dissolved in the ocean mixed layer with respect to the ocean mixed layer depth (%) | | | | | |
|---|---|---|---|---|---|---|---|
| | | 10 - 100 m | 100 - 200 m | 200 - 300 m | 300 - 400 m | 400 - 500 m | 500 - 600 m |
| January | 0.0169 | 94.9 | 2.9 | 1.0 | 0.5 | 0.3 | 0.3 |
| February | 0.0201 | 92.1 | 3.6 | 2.9 | 1.0 | 0.3 | 0.0 |
| March | 0.0255 | 87.8 | 9.2 | 1.7 | 0.7 | 0.2 | 0.4 |
| April | 0.0338 | 66.5 | 31.8 | 1.1 | 0.2 | 0.1 | 0.2 |
| May | 0.0409 | 48.5 | 48.1 | 2.2 | 0.8 | 0.3 | 0.0 |
| June | 0.0510 | 26.8 | 62.7 | 8.0 | 1.7 | 0.8 | 0.1 |
| July | 0.0571 | 14.1 | 69.3 | 12.2 | 3.3 | 0.9 | 0.1 |
| August | 0.0640 | 8.5 | 65.8 | 17.0 | 6.2 | 2.3 | 0.2 |
| September | 0.0609 | 13.5 | 61.0 | 14.6 | 8.2 | 2.7 | 0.0 |
| October | 0.0504 | 24.7 | 58.6 | 12.1 | 2.9 | 1.4 | 0.3 |
| November | 0.0335 | 60.4 | 30.5 | 4.6 | 2.2 | 2.3 | 0.1 |
| December | 0.0196 | 95.1 | 4.3 | 0.4 | 0.2 | 0.0 | 0.0 |

**Table S5. Monthly amount of CH₂F₂ dissolved in the ocean mixed layer at solubility equilibrium with the atmospheric CH₂F₂ (partial pressure, 1 patm) and the depth distribution of the CH₂F₂ dissolved in the southern semi-hemisphere (30°S - 0°S).**

| | Amount (Gg patm$^{-1}$) | Distribution of the amount of CH₂F₂ dissolved in the ocean mixed layer with respect to the ocean mixed layer depth (%) | | | | | |
|---|---|---|---|---|---|---|---|
| | | 10 - 100 m | 100 - 200 m | 200 - 300 m | 300 - 400 m | 400 - 500 m | 500 - 600 m |
| January | 0.0084 | 99.6 | 0.4 | 0 | 0 | 0 | 0 |
| February | 0.0084 | 99.7 | 0.3 | 0 | 0 | 0 | 0 |
| March | 0.0089 | 100.0 | 0 | 0 | 0 | 0 | 0 |
| April | 0.0106 | 100.0 | 0 | 0 | 0 | 0 | 0 |
| May | 0.0131 | 100.0 | 0 | 0 | 0 | 0 | 0 |
| June | 0.0163 | 97.1 | 2.9 | 0 | 0 | 0 | 0 |
| July | 0.0189 | 80.1 | 19.9 | 0 | 0 | 0 | 0 |
| August | 0.0193 | 73.1 | 26.9 | 0 | 0 | 0 | 0 |
| September | 0.0165 | 82.2 | 17.8 | 0 | 0 | 0 | 0 |
| October | 0.0124 | 94.6 | 5.4 | 0 | 0 | 0 | 0 |
| November | 0.0097 | 99.9 | 0.1 | 0 | 0 | 0 | 0 |
| December | 0.0087 | 100.0 | 0 | 0 | 0 | 0 | 0 |

5  **Table S6. Monthly amount of CH₂F₂ dissolved in the ocean mixed layer at solubility equilibrium with the atmospheric CH₂F₂ (partial pressure, 1 patm) and the depth distribution of the CH₂F₂ dissolved in the northern semi-hemisphere (0°N - 30°N).**

| | Amount (Gg patm$^{-1}$) | Distribution of the amount of CH₂F₂ dissolved in the ocean mixed layer with respect to the ocean mixed layer depth (%) | | | | | |
|---|---|---|---|---|---|---|---|
| | | 10 - 100 m | 100 - 200 m | 200 - 300 m | 300 - 400 m | 400 - 500 m | 500 - 600 m |
| January | 0.0132 | 96.4 | 3.6 | 0 | 0 | 0 | 0 |
| February | 0.0126 | 95.9 | 4.1 | 0 | 0 | 0 | 0 |
| March | 0.0107 | 98.7 | 1.3 | 0 | 0 | 0 | 0 |
| April | 0.0087 | 99.8 | 0.2 | 0 | 0 | 0 | 0 |
| May | 0.0079 | 100.0 | 0 | 0 | 0 | 0 | 0 |
| June | 0.0080 | 100.0 | 0 | 0 | 0 | 0 | 0 |
| July | 0.0084 | 100.0 | 0 | 0 | 0 | 0 | 0 |
| August | 0.0082 | 100.0 | 0 | 0 | 0 | 0 | 0 |
| September | 0.0080 | 100.0 | 0 | 0 | 0 | 0 | 0 |
| October | 0.0086 | 100.0 | 0 | 0 | 0 | 0 | 0 |
| November | 0.0100 | 100.0 | 0 | 0 | 0 | 0 | 0 |
| December | 0.0118 | 100.0 | 0 | 0 | 0 | 0 | 0 |

**Table S7. Monthly amount of $CH_2F_2$ dissolved in the ocean mixed layer at solubility equilibrium with the atmospheric $CH_2F_2$ (partial pressure, 1 patm) and the depth distribution of the $CH_2F_2$ dissolved in the northern semi-hemisphere (30°N - 90°N).**

| | Amount (Gg patm$^{-1}$) | Distribution of the amount of $CH_2F_2$ dissolved in the ocean mixed layer with respect to the ocean mixed layer depth (%) | | | | | |
|---|---|---|---|---|---|---|---|
| | | 10 - 100 m | 100 - 200 m | 200 - 300 m | 300 - 400 m | 400 - 500 m | 500 - 600 m |
| January | 0.0205 | 41.3 | 50.1 | 7.0 | 1.4 | 0.2 | 0.0 |
| February | 0.0225 | 34.5 | 55.3 | 7.1 | 2.3 | 0.6 | 0.2 |
| March | 0.0208 | 49.7 | 42.3 | 4.9 | 1.7 | 0.7 | 0.6 |
| April | 0.0147 | 79.7 | 17.6 | 1.7 | 0.4 | 0.0 | 0.6 |
| May | 0.0081 | 90.1 | 9.9 | 0 | 0 | 0 | 0 |
| June | 0.0055 | 97.7 | 2.3 | 0 | 0 | 0 | 0 |
| July | 0.0045 | 96.6 | 3.4 | 0 | 0 | 0 | 0 |
| August | 0.0048 | 94.4 | 5.6 | 0 | 0 | 0 | 0 |
| September | 0.0059 | 97.7 | 2.3 | 0 | 0 | 0 | 0 |
| October | 0.0084 | 99.6 | 0.4 | 0 | 0 | 0 | 0 |
| November | 0.0121 | 89.6 | 10.4 | 0.1 | 0 | 0 | 0 |
| December | 0.0163 | 71.0 | 26.1 | 2.9 | 0 | 0 | 0 |

**R2-7** -----------------------------------------------------------------------------------------------------------------

(1) comments from Referee 2

The conclusion that 5% of the atmospheric burden of $CH_2F_2$ would reside in the ocean mixed layer in the southern semi-hemispheric lower troposphere during winter seems to be an upper limit, and should be worded as such. How much lower could this upper limit be?

(2) author's response

Thank you for the comments. As described in R2-6, it takes days to a few weeks after a change in temperature or salinity for oceanic surface mixed layers to come to equilibrium with the present atmosphere, and equilibration time increases with depth of the surface mixed layer (Fine, 2011).

Haine and Richards (1995) demonstrated that the seasonal variation in ocean mixed layer depth was the key process which affected undersaturation and supersaturation of chlorofluorocarbon 11 (CFC-11), CFC-12 and CFC-113 by use of a one-dimensional slab mixed model. Specifically, the mixed layer deepening in autumn would cause undersaturation in the mixed layer. In the estimation, >90 % of $CH_2F_2$ in the ocean mixed layer is expected to reside in less than 300 m depth (Tables S4, S5, S6 and S7). According to the report by Haine and Richards (1995), saturation of $CH_2F_2$ would be >0.9 for the ocean mixed layer with less than 300 m depth. The saturation of $CH_2F_2$ in the ocean mixed layer is thus estimated to be at least 0.8.

The manuscript will be revised, as described in R2-6, and Fine (2011) and Haine and Richards (1995) will be cited.

(3) author's changes in manuscript

5   lines 13-16, page 1, Abstract

By using this equation in a lower tropospheric semi-hemisphere (30° S−90° S) of the Advanced Global Atmospheric Gases Experiment (AGAGE) 12-box model, we estimated that 1 to 4 % of the atmospheric burden of $CH_2F_2$ resided in the ocean mixed layer and that this percentage was at least 4 % in the winter; dissolution of $CH_2F_2$ in the ocean may partially influence estimates of $CH_2F_2$ emissions from long-term observational data of atmospheric $CH_2F_2$ concentrations.

lines 20-24, page 12

By using the solubility of $CH_2F_2$ determined in this study, the magnitude of buffering of the atmospheric burden of $CH_2F_2$ by the additional $CH_2F_2$ in ocean surface waters is estimated to be realistically limited to only about 1 % globally; however, in a southern semi-hemispheric lower troposphere (30° S–90° S) of the AGAGE 12-box model, the atmospheric burden of

15   $CH_2F_2$ is estimated to reside in the ocean mixed layer by at least 4 % in the winter and by 1 % in the summer.

Other changes are included in the change mentioned in R2-6.

**R2-8** --------------------------------------------------------------------------------------------------------------------

20   (1) comments from Referee 2

It seems surprising that the dissolution of $CH_2F_2$ into the ocean should affect estimates of $CH_2F_2$ emissions in the Southern Hemisphere and their seasonal variability, because the atmospheric concentrations that reach the Southern Hemisphere are also affected by transport, and chemical removal, and related uncertainties. This should be mentioned.

(2) author's response

Thank you for the comment. As Referee 2 pointed out, the atmospheric concentrations that reach the Southern Hemisphere are also affected by transport, chemical removal, and related uncertainties; this should be mentioned.

I will first describe how the dissolution of $CH_2F_2$ into the ocean may affect estimation of $CH_2F_2$ emissions in the

30   Southern Hemisphere and their seasonal variability, and then I will show the revised text.

In 2012, atmospheric concentrations of $CH_2F_2$ in the Northern Hemisphere are by >30% higher than in the Southern Hemisphere (O'Doherty et al., 2014); the strong inter-hemisphere gradient indicates that emissions of $CH_2F_2$ are predominantly in the Northern Hemisphere. In the AGAGE 12 box model (Rigby et al., 2013), transport of $CH_2F_2$ is dominated by eddy diffusion between the boxes in the model. The seasonal eddy diffusion

parameters between the Northern Hemisphere and the Southern Hemisphere in the model are 187 to 568 days in lower troposphere, and 81 to 109 days in upper troposphere (Rigby et al., 2013).

The rate of increase in atmospheric concentration of $CH_2F_2$ due to the emission of $CH_2F_2$ in the Southern Hemisphere, which is denoted as $RE_{south}$ hereafter, is thus more sensitive to change in atmospheric
5    concentrations of $CH_2F_2$ in the Southern Hemisphere than those in the Northern Hemisphere, partly because $CH_2F_2$ is removed through gas phase reactions with OH (partial atmospheric lifetime of 5.5 years). Furthermore, $RE_{south}$ would range small values such as a few % $y^{-1}$ or less because emissions of $CH_2F_2$ were predominantly in the Northern Hemisphere and because, in 2012, the rate of increase in the global mean mole fraction of $CH_2F_2$ was 17% $y^{-1}$ (O'Doherty et al., 2014). In estimation of $RE_{south}$, small value of $RE_{south}$ would be deduced from
10    difference in the rates of increase of atmospheric concentrations of $CH_2F_2$ between hemispheres. Dissolution of $CH_2F_2$ in the ocean in the Southern Hemisphere may thus affect estimation of $RE_{south}$ and then affect estimation of $CH_2F_2$ emissions in the Southern Hemisphere and their seasonal variability.

I revise the text as follows.

15    (3) author's changes in manuscript
lines 12-16, page 12

In the Southern Hemisphere, $CH_2F_2$ emission rates are much lower than in the Northern Hemisphere. Hence, dissolution of $CH_2F_2$ in the ocean, even if dissolution is reversible, may influence estimates of $CH_2F_2$ emissions derived from long-term observational data on atmospheric concentrations of $CH_2F_2$; in particular, consideration of dissolution of $CH_2F_2$ in the ocean
20    may affect estimates of $CH_2F_2$ emissions in the Southern Hemisphere and their seasonal variability because of slow rates of inter-hemispheric transport and small portion of the $CH_2F_2$ emissions in the Southern Hemisphere to the total emissions.

**R2-9** ------------------------------------------------------------------------------------------------------------------------------

(1) comments from Referee 2
25    On line 27, page 2 (in the revised manuscript), 'first' is written twice.

(2) author's response

Thank you for the comment. The text is revised.

30    (3) author's changes in manuscript
line 27, page 2

First, the values of $K_H$ for $CH_2F_2$ were determined over the temperature range from 276 to 313 K by means of an inert-gas stripping (IGS) method.

**R2-10----------------------------------------------------------------------------------------------------------------------**

(1) comments from Referee 2

   On lines 9-10, page 5 (in the revised manuscript), add errors for numbers. See comments #1, #2 and add typical values, their units, and uncertainties of variables for the key equations throughout the manuscript.

(2) author's response

   Thank you for the comment. If redistribution of $CH_2F_2$ in the headspace to the test solution had occurred, the $K_{eq}$ values determined in this study would be overestimated. Errors due to this redistribution are always negative values. The ratio of the errors to the $K_{eq}$ values (%) is $100 \times \frac{\left(\frac{V_{head}}{RTV}\right)}{\left(\frac{1}{k_1 RTV}F\right)}$, that is, $\frac{100 k_1 V_{head}}{F}$. Under the experimental

10   conditions here, this ratio is calculated to be −2.0 to −2.3 % at 3.0 °C and −4.6 to −5.1 % at 39.5 °C. Values of this ratio increase as values of $K_{eq}$ decrease. This ratio is maximum (−6.5 %) for a-seawater at 51.534‰ and 39.5 °C.

   Typical values, their units, and uncertainties of variables for the key equations are added in the revised manuscript.

(3) author's changes in manuscript

lines 6-8, page 4

[revised manuscript text omitted]

**R2-11**------------------------------------------------------------------------------------------------------------------------------

(1) comments from Referee 2

What statistical test for outliers was applied? How many points were removed at each temperature?

(2) author's response

Thank you for the comment. Statistical test for outliers is as follows.

The data with errors being >10% of the data was first excluded. Next, some data were excluded for calculation of the average so that the remaining data were inside the 2σ range. This procedure was iterated until all the data were inside the 2σ range.

The number of them were eight or fewer at each temperature. The maximum number of the data excluded was corrected to be eight although it was described to be six in the original manuscript. Number of the data thus excluded were indicated in Tables 1 and 3 (R1-1).

(3) author's changes in manuscript

lines 19-22, page 7

The data with errors being >10% of the data was first excluded. Next, some data were excluded for calculation of the

average so that the remaining data were inside the 2σ range. This procedure was iterated until all the data were inside the 2σ range. The data points thus excluded was only for $V$ values of 0.350 dm$^3$ and the number of them were eight or fewer at each temperature.

5  Tables 1 and 3

These tables in the revised manuscript are shown in R1-1.

**R2-12**------------------------------------------------------------------------------------------------------------------------------------------------

(1) comments from Referee 2

10  About Eq (17), for sake of discussion, can a $k_S$ value be given here? And what is the effect of including $k_S$ vs $k_{S1}$, $k_{s2}$ in the model - does it make a difference?

(2) author's response

Thank you for the comment. Parameters of $k_{S1}$, $k_{s2}$ in the original manuscript are replaced by a parameter of 15  $k_{S1}$ in the revised manuscript. I add Table S2. Table S2 lists values of $k_s$ and comparison of the $K_{eq}$ values calculated between by Eq. (17) and by Eq. (22) at salinity of 30, 35 and 40 ‰ and each temperature.

(3) author's changes in manuscript

lines 27-30, page 9

20  Figure 5 plots $\ln(K_H(T)/K_{eq}{}^S(T))$ against $S$ at each temperature. Table S2 lists values of $k_s$ determined by fitting the data at each temperature by use of Eq. (17). If the $K_{eq}{}^S(T)$ values obeyed Eq. (17), the data at each temperature in Fig. 5 would fall on a straight line passing through the origin, but they did not. Figure 5 reveals that the salinity dependence of $CH_2F_2$ solubility in a-seawater cannot be represented by Eq. (17).

25  lines 18-21, page 10

$\ln(K_H(T)/K_{eq}{}^S(T))$ depends on $S^{0.5}$ and follows Eq. (22) rather than the Sechenov equation (Eq. (17)). Table S2 compares values of $K_{eq}{}^S$ calculated by Eq. (22) with those by Eq. (17). The difference between these values of $K_{eq}{}^S$ at 35‰ of salinity was within 3% of the $K_{eq}{}^S$ value. Decreases in values of $K_{eq}{}^S$ are calculated to be 7–8% and 4%, respectively, by Eqs. (17) and (22) as salinity of a-seawater increases from 30‰ to 40‰ at each temperature.

Table S2

**Table S2. Values of $k_s$ (Eq. (17)) and comparison of values of $K_{eq}{}^S$ calculated at each temperature by Eq. (17) with those by Eq. (22).**

[revised manuscript text omitted]

a. Errors are 2σ for the regression only.; b. Errors are those at 95% confidence level for the regression only.; c. Number in parenthesis represents both errors at 95% confidence level for the regression and potential systematic bias (±4%).

[Figure]

**Figure S3. Headspace GC-MS measurements for six series of test samples containing water ($V_i$ in cm$^3$) to which a CH$_2$F$_2$–air mixture was added ($v_j$ in cm$^3$) at 313 K. (a) Plot of peak area ($S_{ij}$) versus $v_j$ for test samples containing volume $V_i$ of water. Slope ($L_i$) was obtained by linear fitting of the data to Eq. (8) for samples of the same $V_i$. (b) Plot of $L_i^{-1}$ versus $V_i/V_0$ fitted to Eq. (10).**

[Figure]

**Figure S4. Plot of $L_i$ versus $V_i/V_0$ for the PRV-HS measurements at each temperature. Bold curves represent the simultaneous fitting of the two datasets at each temperature by Eq. (11).**

**S4. Determination of salting-out effects in artificial seawater**

[Figure]

**Figure S5. Plots of values of $F/(k_1RTV)$ against $F$ at each temperature for 0.35 dm³ of a-seawater at 4.452‰. Error bars represent 2σ due to errors of values of $k_1$ as described in Sect. S2. Grey symbols represent the data excluded for calculating the average.**

[Figure]

**Figure S6. Plots of values of $F/(k_1RTV)$ against $F$ at each temperature for 0.35 dm³ of a-seawater at 8.921‰. Error bars represent 2σ due to errors of values of $k_1$ as described in Sect. S2. Grey symbols represent the data excluded for calculating the average.**

[Figure]

**Figure S7. Plots of values of $F/(k_1RTV)$ against $F$ at each temperature for 0.35 dm³ of a-seawater at 21.520‰. Error bars represent 2σ due to errors of values of $k_1$ as described in Sect. S2. Grey symbols represent the data excluded for calculating the average.**

[Figure]

**Figure S8. Plots of values of $F/(k_1RTV)$ against $F$ at each temperature for 0.35 dm³ of a-seawater at 51.534‰. Error bars represent 2σ due to errors of values of $k_1$ as described in Sect. S2. Grey symbols represent the data excluded for calculating the average.**

[Figure]

**Figure S9. log-log plots for $\ln(K_H(T)/K_H^S K_{eq}^S(T))$ vs. salinity in a-seawater at each temperature. Bold lines represent the fitting obtained by a liner regression. Errors are those at 95% confidence level for the regression only.**

[Figure]

**Figure S10. Plots of $k_{s1}$ and $k_{s2}$ (coefficients in Eq. (18)) against temperature.**

**Table S2. Values of $k_s$ (Eq. (17)) and comparison of values of $K_{eq}{}^S$ calculated at each temperature by Eq. (17) with those by Eq. (22).**

| Temperature (°C) | $k_s$ (‰$^{-1}$) | [$K_{eq}{}^S$ from Eq. (17)]/ [$K_{eq}{}^S$ from Eq. (22)] | | | [$K_{eq}{}^S$ at 30‰]/ [$K_{eq}{}^S$ at 40‰] | |
|---|---|---|---|---|---|---|
| | | at 30‰ | at 35‰ | at 40‰ | Eq. (17) | Eq. (22) |
| 3.0 | 0.00811 | 1.027 | 1.008 | 0.988 | 1.084 | 1.043 |
| 5.8 | 0.00785 | 1.033 | 1.014 | 0.995 | 1.082 | 1.042 |
| 10.5 | 0.00768 | 1.033 | 1.016 | 0.997 | 1.080 | 1.042 |
| 15.5 | 0.00718 | 1.044 | 1.028 | 1.012 | 1.074 | 1.041 |
| 20.3 | 0.00728 | 1.037 | 1.020 | 1.003 | 1.076 | 1.040 |
| 25.0 | 0.00704 | 1.040 | 1.024 | 1.008 | 1.073 | 1.039 |
| 29.9 | 0.00731 | 1.027 | 1.010 | 0.992 | 1.076 | 1.039 |
| 34.8 | 0.00713 | 1.029 | 1.012 | 0.995 | 1.074 | 1.038 |
| 39.5 | 0.00709 | 1.026 | 1.010 | 0.992 | 1.073 | 1.038 |

**S5. Discussion of potential reason for this salting-out effect of $CH_2F_2$ solubility in a-seawater (deviation from Sechenov relationship)**

The reason that the salting-out effect of $CH_2F_2$ solubility in a-seawater depends on $S^{0.5}$ is not clear. Specific properties of $CH_2F_2$ –small molecular volume, which results in small work of cavity creation (Graziano, 2004; 2008), and large solute-solvent attractive potential energy in water and a-seawater– may cause deviation from Sechenov relationship. This possibility may be discussed here.

I calculate Ben-Naim standard Gibbs energy $\Delta G^{\bullet}$, enthalpy $\Delta H^{\bullet}$, and entropy $\Delta S^{\bullet}$ changes for dissolution of $CH_2F_2$ in water because these values correspond to the values for the transfer from a fixed position in the gas phase to a fixed position in water. Values of $\Delta G^{\bullet}$, $\Delta H^{\bullet}$, and $\Delta S^{\bullet}$ are calculated on the basis of the Ostwald solubility coefficient, $L(T)$, as follows.

$$\ln(L(T)) = \ln\left(RTK_{eq}{}^{S}(T)\right) \tag{B1}$$

$$\Delta G^{\bullet} = R'T\ln(L(T)) \tag{B2}$$

$$\Delta H^{\bullet} = -\frac{\partial}{\partial(^1/_T)}\left(\frac{\Delta G^{\bullet}}{T}\right) \tag{B3}$$

$$\Delta S^{\bullet} = \frac{\Delta H^{\bullet} - \Delta G^{\bullet}}{T} \tag{B4}$$

where both $R$ and $R'$ represent gas constant but their units are different: $R$ = 0.0821 in atm dm$^3$ K$^{-1}$ mol$^{-1}$; $R'$ = 8.314 in J K$^{-1}$ mol$^{-1}$.

Combining Eqs. (B1), (B2), (B3), and (B4) with Eqs. (14) and (15), $\Delta G^{\bullet}$ (kJ mol$^{-1}$), $\Delta H^{\bullet}$ (kJ mol$^{-1}$), and $\Delta S^{\bullet}$ (J mol$^{-1}$ K$^{-1}$) are represented by $\Delta G_{sol}$ and $\Delta H_{sol}$ as follows:

$$\Delta G^{\bullet} = \Delta G_{sol} + R'T\ln(RT) \tag{B5}$$

$$\Delta H^{\bullet} = \Delta H_{sol} + R'T \tag{B6}$$

$$\Delta S^{\bullet} = \frac{\Delta H_{sol} - \Delta G_{sol}}{T} + R' - R'\ln(RT) \tag{B7}$$

Values of $\Delta G^{\bullet}$, $\Delta H^{\bullet}$, and $\Delta S^{\bullet}$ calculated at 298 K are listed in Table S3. Table S3 also lists values of $\Delta G^{\bullet}$, $\Delta H^{\bullet}$, and $\Delta S^{\bullet}$ reported for $CH_3F$ and $C_2H_6$ (Graziano, 2004) and $CH_4$ (Graziano, 2008) at 298 K. The chemicals, which having a methyl group, in Table S3 are classified into two groups ($CH_2F_2$ and $CH_3F$; $CH_4$ and $C_2H_6$) according to $\Delta G^{\bullet}$.

Table S3 lists values of $\Delta G_c$, $E_a$ and $\Delta H^h$ deduced using a scaled particle theory (Granziano, 2004; 2008). $\Delta G_c$ is the work of cavity creation to insert a solute in a solvent. $E_a$ is a solute-solvent attractive potential energy and accounts for the solute-solvent interactions consisting of dispersion, dipole-induced dipole, and dipole-dipole contributions. $\Delta H^h$ is enthalpy of solvent molecules reorganization caused by solute insertion. The solvent reorganization mainly involves a rearrangement of H-bonds.

$\Delta G_c$ is entropic in nature in all liquids, being a measure of the excluded volume effect due to a reduction in the spatial configurations accessible to liquid molecules upon cavity creation. Hence, $C_2H_6$ has larger value of $\Delta G_c$ than $CH_3F$ and $CH_4$. $\Delta G_c$, $E_a$, and $\Delta H^h$ are related to $\Delta G^{\bullet}$ and $\Delta H^{\bullet}$ as follows (Graziano, 2008):

$$\Delta G^{\cdot} = \Delta G_c + E_a \quad\quad\quad\quad\quad\quad\quad\quad\quad\quad\quad\quad\quad\quad\quad\quad\quad\quad\quad\quad\quad\text{(B8)}$$

$$\Delta H^{\cdot} = E_a + \Delta H^h \quad\quad\quad\quad\quad\quad\quad\quad\quad\quad\quad\quad\quad\quad\quad\quad\quad\quad\quad\quad\text{(B9)}$$

Table S3 thus suggests that smaller value of $\Delta G^{\cdot}$ of $CH_3F$ than $CH_4$ is due to large solute-solvent attractive potential energy ($-E_a$) of $CH_3F$.

**Table S3. Ben-Naim standard hydration Gibbs energy $\Delta G^{\cdot}$, enthalpy $\Delta H^{\cdot}$, and entropy $\Delta S^{\cdot}$ changes for dissolution of $CH_2F_2$ at 298 K determined here and the corresponding values and values of $\Delta G_c$, $E_a$ and $\Delta H^h$ reported for $CH_3F$ and $C_2H_6$ (Granziano, 2004) and $CH_4$ (Graziano, 2008).**

| | $\Delta G^{\cdot}$ (kJ mol$^{-1}$) | $\Delta H^{\cdot}$ (kJ mol$^{-1}$) | $\Delta S^{\cdot}$ (J K$^{-1}$ mol$^{-1}$) | $\Delta G_c$ (kJ mol$^{-1}$) | $E_a$ (kJ mol$^{-1}$) | $\Delta H^h$ (kJ mol$^{-1}$) |
|---|---|---|---|---|---|---|
| $CH_2F_2$ | −1.1 | −14.7 | −45.4 | | | |
| $CH_3F$ | −0.9 | −15.8 | −50.0 | 23.3 | −24.3 | 8.5 |
| $CH_4$ | 8.4 | −10.9 | −64.7 | 22.9 | −14.5 | 3.7 |
| $C_2H_6$ | 7.7 | −17.5 | −84.5 | 28.4 | −20.7 | 3.2 |

Graziano (2008) definitively explained the salting-out of $CH_4$ by sodium chloride at molecular level on the basis of a scaled particle theory. He explained that $\Delta G_c$ increase was linearly related to the increase in the volume packing density of the solutions ($\xi_3$) with adding NaCl. Such an increase of $\Delta G_c$ is probably the case for salting-out of $CH_2F_2$ by a-seawater observed in this study. He also explained that $E_a$ was linearly related to the increase in $\xi_3$ assuming that a fraction of the dipole-induced dipole attractions could be taken into account by the parameterization of the dispersion contribution.

I think the possibility that $E_a$ may be nonlinearly related to the increase in $\xi_3$ because of dipole-dipole interaction between $CH_2F_2$ and solvents. Temperature dependence in Eq. (22) suggests that salting-out effect of $CH_2F_2$ by a-seawater is enthalpic. Eqs. (22) and (B9) thus suggests that the salting-out of $CH_2F_2$ is mostly related to change in $E_a$. $CH_2F_2$ has relatively small value of $\Delta G_c$ because of its small molecular volume compared to other chemicals such as $C_2H_6$. Accordingly, $\Delta G^{\cdot}$, that is, solubility of $CH_2F_2$ would depend on $E_a$ rather than $\Delta G_c$. Therefore, I think that specific properties of $CH_2F_2$ – small molecular volume, which results in small work of cavity creation (Graziano, 2004; 2008), and large solute-solvent attractive potential energy in water and a-seawater− may cause deviation from Sechenov relationship.

**Table S4. Monthly amount of CH$_2$F$_2$ dissolved in the ocean mixed layer at solubility equilibrium with the atmospheric CH$_2$F$_2$ (partial pressure, 1 patm) and the depth distribution of the CH$_2$F$_2$ dissolved in the southern semi-hemisphere (90° S - 30° S).**

| | Amount (Gg patm$^{-1}$) | Distribution of the amount of CH$_2$F$_2$ dissolved in the ocean mixed layer with respect to the ocean mixed layer depth (%) | | | | | |
|---|---|---|---|---|---|---|---|
| | | 10 - 100 m | 100 - 200 m | 200 - 300 m | 300 - 400 m | 400 - 500 m | 500 - 600 m |
| January | 0.0169 | 94.9 | 2.9 | 1.0 | 0.5 | 0.3 | 0.3 |
| February | 0.0201 | 92.1 | 3.6 | 2.9 | 1.0 | 0.3 | 0.0 |
| March | 0.0255 | 87.8 | 9.2 | 1.7 | 0.7 | 0.2 | 0.4 |
| April | 0.0338 | 66.5 | 31.8 | 1.1 | 0.2 | 0.1 | 0.2 |
| May | 0.0409 | 48.5 | 48.1 | 2.2 | 0.8 | 0.3 | 0.0 |
| June | 0.0510 | 26.8 | 62.7 | 8.0 | 1.7 | 0.8 | 0.1 |
| July | 0.0571 | 14.1 | 69.3 | 12.2 | 3.3 | 0.9 | 0.1 |
| August | 0.0640 | 8.5 | 65.8 | 17.0 | 6.2 | 2.3 | 0.2 |
| September | 0.0609 | 13.5 | 61.0 | 14.6 | 8.2 | 2.7 | 0.0 |
| October | 0.0504 | 24.7 | 58.6 | 12.1 | 2.9 | 1.4 | 0.3 |
| November | 0.0335 | 60.4 | 30.5 | 4.6 | 2.2 | 2.3 | 0.1 |
| December | 0.0196 | 95.1 | 4.3 | 0.4 | 0.2 | 0.0 | 0.0 |

**Table S5. Monthly amount of CH$_2$F$_2$ dissolved in the ocean mixed layer at solubility equilibrium with the atmospheric CH$_2$F$_2$ (partial pressure, 1 patm) and the depth distribution of the CH$_2$F$_2$ dissolved in the southern semi-hemisphere (30° S - 0° S).**

| | Amount (Gg patm$^{-1}$) | Distribution of the amount of CH$_2$F$_2$ dissolved in the ocean mixed layer with respect to the ocean mixed layer depth (%) | | | | | |
|---|---|---|---|---|---|---|---|
| | | 10 - 100 m | 100 - 200 m | 200 - 300 m | 300 - 400 m | 400 - 500 m | 500 - 600 m |
| January | 0.0084 | 99.6 | 0.4 | 0 | 0 | 0 | 0 |
| February | 0.0084 | 99.7 | 0.3 | 0 | 0 | 0 | 0 |
| March | 0.0089 | 100.0 | 0 | 0 | 0 | 0 | 0 |
| April | 0.0106 | 100.0 | 0 | 0 | 0 | 0 | 0 |
| May | 0.0131 | 100.0 | 0 | 0 | 0 | 0 | 0 |
| June | 0.0163 | 97.1 | 2.9 | 0 | 0 | 0 | 0 |
| July | 0.0189 | 80.1 | 19.9 | 0 | 0 | 0 | 0 |
| August | 0.0193 | 73.1 | 26.9 | 0 | 0 | 0 | 0 |
| September | 0.0165 | 82.2 | 17.8 | 0 | 0 | 0 | 0 |
| October | 0.0124 | 94.6 | 5.4 | 0 | 0 | 0 | 0 |
| November | 0.0097 | 99.9 | 0.1 | 0 | 0 | 0 | 0 |
| December | 0.0087 | 100.0 | 0 | 0 | 0 | 0 | 0 |

**Table S6. Monthly amount of CH$_2$F$_2$ dissolved in the ocean mixed layer at solubility equilibrium with the atmospheric CH$_2$F$_2$ (partial pressure, 1 patm) and the depth distribution of the CH$_2$F$_2$ dissolved in the northern semi-hemisphere (0° N - 30° N).**

| | Amount (Gg patm$^{-1}$) | Distribution of the amount of CH$_2$F$_2$ dissolved in the ocean mixed layer with respect to the ocean mixed layer depth (%) | | | | | |
|---|---|---|---|---|---|---|---|
| | | 10 - 100 m | 100 - 200 m | 200 - 300 m | 300 - 400 m | 400 - 500 m | 500 - 600 m |
| January | 0.0132 | 96.4 | 3.6 | 0 | 0 | 0 | 0 |
| February | 0.0126 | 95.9 | 4.1 | 0 | 0 | 0 | 0 |
| March | 0.0107 | 98.7 | 1.3 | 0 | 0 | 0 | 0 |
| April | 0.0087 | 99.8 | 0.2 | 0 | 0 | 0 | 0 |
| May | 0.0079 | 100.0 | 0 | 0 | 0 | 0 | 0 |
| June | 0.0080 | 100.0 | 0 | 0 | 0 | 0 | 0 |
| July | 0.0084 | 100.0 | 0 | 0 | 0 | 0 | 0 |
| August | 0.0082 | 100.0 | 0 | 0 | 0 | 0 | 0 |
| September | 0.0080 | 100.0 | 0 | 0 | 0 | 0 | 0 |
| October | 0.0086 | 100.0 | 0 | 0 | 0 | 0 | 0 |
| November | 0.0100 | 100.0 | 0 | 0 | 0 | 0 | 0 |
| December | 0.0118 | 100.0 | 0 | 0 | 0 | 0 | 0 |

**Table S7. Monthly amount of CH$_2$F$_2$ dissolved in the ocean mixed layer at solubility equilibrium with the atmospheric CH$_2$F$_2$ (partial pressure, 1 patm) and the depth distribution of the CH$_2$F$_2$ dissolved in the northern semi-hemisphere (30° N - 90° N).**

| | Amount (Gg patm$^{-1}$) | Distribution of the amount of CH$_2$F$_2$ dissolved in the ocean mixed layer with respect to the ocean mixed layer depth (%) | | | | | |
|---|---|---|---|---|---|---|---|
| | | 10 - 100 m | 100 - 200 m | 200 - 300 m | 300 - 400 m | 400 - 500 m | 500 - 600 m |
| January | 0.0205 | 41.3 | 50.1 | 7.0 | 1.4 | 0.2 | 0.0 |
| February | 0.0225 | 34.5 | 55.3 | 7.1 | 2.3 | 0.6 | 0.2 |
| March | 0.0208 | 49.7 | 42.3 | 4.9 | 1.7 | 0.7 | 0.6 |
| April | 0.0147 | 79.7 | 17.6 | 1.7 | 0.4 | 0.0 | 0.6 |
| May | 0.0081 | 90.1 | 9.9 | 0 | 0 | 0 | 0 |
| June | 0.0055 | 97.7 | 2.3 | 0 | 0 | 0 | 0 |
| July | 0.0045 | 96.6 | 3.4 | 0 | 0 | 0 | 0 |
| August | 0.0048 | 94.4 | 5.6 | 0 | 0 | 0 | 0 |
| September | 0.0059 | 97.7 | 2.3 | 0 | 0 | 0 | 0 |
| October | 0.0084 | 99.6 | 0.4 | 0 | 0 | 0 | 0 |
| November | 0.0121 | 89.6 | 10.4 | 0.1 | 0 | 0 | 0 |
| December | 0.0163 | 71.0 | 26.1 | 2.9 | 0 | 0 | 0 |